# DIP2 is a unique regulator of diacylglycerol lipid homeostasis in eukaryotes

**Sudipta Mondal**[1†], **Priyadarshan Kinatukara**[1†], **Shubham Singh**[2],
**Sakshi Shambhavi**[1,3], **Gajanan S Patil**[1,3], **Noopur Dubey**[1], **Salam Herojeet Singh**[1],
**Biswajit Pal**[1], **P Chandra Shekar**[1], **Siddhesh S Kamat**[2], **Rajan Sankaranarayanan**[1,3]*

[1]CSIR-Centre for Cellular and Molecular Biology, Hyderabad, India; [2]Department of Biology, Indian Institute of Science Education and Research (IISER), Pune, India; [3]Academy of Scientific and Innovative Research (AcSIR), Ghaziabad, India

**Abstract** Chain-length-specific subsets of diacylglycerol (DAG) lipids are proposed to regulate differential physiological responses ranging from signal transduction to modulation of the membrane properties. However, the mechanism or molecular players regulating the subsets of DAG species remain unknown. Here, we uncover the role of a conserved eukaryotic protein family, DISCO-interacting protein 2 (DIP2) as a homeostatic regulator of a chemically distinct subset of DAGs using yeast, fly, and mouse models. Genetic and chemical screens along with lipidomics analysis in yeast reveal that DIP2 prevents the toxic accumulation of specific DAGs in the logarithmic growth phase, which otherwise leads to endoplasmic reticulum stress. We also show that the fatty acyl-AMP ligase-like domains of DIP2 are essential for the redirection of the flux of DAG subspecies to storage lipid, triacylglycerols. DIP2 is associated with vacuoles through mitochondria–vacuole contact sites and such modulation of selective DAG abundance by DIP2 is found to be crucial for optimal vacuole membrane fusion and consequently osmoadaptation in yeast. Thus, the study illuminates an unprecedented DAG metabolism route and provides new insights on how cell fine-tunes DAG subspecies for cellular homeostasis and environmental adaptation.

**\*For correspondence:**
sankar@ccmb.res.in

[†]These authors contributed
equally to this work

**Competing interest:** See page
24

**Reviewing Editor:** Felix
Campelo, Institute of Photonic
Sciences, Spain

## Editor's evaluation

This manuscript reports that a previously uncharacterized protein, DIP2, which is localized at mitochondria and mitochondria-vacuole contacts is involved in the metabolism of a subset of diacylglycerol species. This study would be of interest to biologists interested in lipids, membrane contacts, metabolism, and ER stress.

## Introduction

Eukaryotic cells are functionally compartmentalized by membranes and the integrity of these compartments requires non-uniform distribution of different lipids (*van Meer et al., 2008*). The physicochemical properties of different lipids are attributed to the bilayer formation, regulation of membrane protein activity, membrane curvature, and fusion–fission processes (*Bigay and Antonny, 2012*; *Harayama and Riezman, 2018*). Diacylglycerol (DAG) is one such crucial bioactive lipid functioning at the crossroads of cell signalling and core metabolic pathways including membrane biogenesis and fat storage. Intriguingly, a growing body of evidence suggests that chemically defined subsets of DAGs, with specific fatty acid (FA) tails, determine the fate of fundamental cellular processes (*Marignani et al., 1996*; *Schuhmacher et al., 2020*; *Ware et al., 2020*). Therefore, precise regulation of DAG

**eLife digest** Lipids, such as fats and hormones, constitute one of the main building blocks of cells. There are thousands of different lipids each with distinctive chemical properties that allow them to carry out specific roles. For example, a group of lipids called diacylglycerols help cells perform a myriad of tasks, like sensing external signals, making membranes, and storing energy. The production and breakdown of diacylglycerols is therefore tightly regulated. However, very little is known about the molecules involved in this metabolic process.

One possible candidate is the enzyme DIP2 which is comprised of a protein module known as FAAL (short for fatty acyl-AMP ligase). FAAL belongs to a family of enzymes that synthesize lipid-like molecules in bacteria. In 2021, a group of researchers tracked the evolutionary trajectory of these bacterial proteins and found that most of them were lost in eukaryotes, such as animals and fungi. FAAL-like proteins, however, had been retained through evolution and incorporated in to DIP2.

Here, Mondal, Kinatukara et al. – including some of the researchers involved in the 2021 study – have used a combination of genetic and biochemical experiments to investigate whether and how DIP2 contributes to lipid metabolism in eukaryotes. They found that yeast cells without the gene for DIP2 had higher levels of diacylglycerols which hampered the shape and function of certain cellular compartments. The mutant cells were also unable to convert diacylglycerols in to another group of lipids which are involved in energy storage. This effect was observed in fruit flies and mice lacking DIP2, suggesting that this role for DIP2 is conserved across most eukaryotes.

Further experiments in yeast cells revealed that unlike other enzymes that metabolize diacylglycerols, DIP2 only acted on a sub-population of diacylglycerols at specific locations and times. Furthermore, yeast cells lacking DIP2 could still grow under ideal conditions, but could not cope with high or low salt concentrations in their surroundings, suggesting that the enzyme helps cells deal with environmental stresses.

Since DIP2 is found in most eukaryotes, understanding how it works could be useful for multiple branches of biology. For example, some pathogenic fungi that cause diseases in crop plants and humans also rely on DIP2. Further studies are needed to better understand the role that DIP2 plays in other eukaryotic species which may shed light on other processes the enzyme is involved in.

subspecies through selective formation and degradation is crucial while the identity of molecular players and their mechanisms remain unexplored.

A key step of lipid metabolism is the incorporation of FA of different chain lengths and unsaturation in lipids during biosynthesis or remodelling (*Grevengoed et al., 2014*; *Watkins, 1997*). In either case, FA is activated via adenylation reaction using ATP molecules by fatty acyl-CoA ligases (FACLs), a member of the ANL superfamily of enzymes (*Gulick, 2009*). However, a recently identified homologue of FACLs, named fatty acyl-AMP ligases (FAALs) also activate FA but redirect them to 4'-phosphopantetheine coupled to acyl-carrier protein (ACP) in bacteria. The complex lipids of Mycobacteria, lipopeptides in several bacteria are the classical examples of such metabolic diversions created by FAALs, where they crosstalk with polyketide synthases (PKS) and non-ribosomal peptide synthetases (NRPS) (*Arora et al., 2009*; *Goyal et al., 2012*; *Trivedi et al., 2004*). We have recently identified a distant orthologue of FAALs as a part of a conserved three-domain protein called DIP2 across the eukaryotic supergroup Opisthokonta (fungi and animals) (*Patil et al., 2021*). However, paradoxically, most of the opisthokonts have lost PKS/NRPS gene cluster, suggesting the emergence of possible alternate functions of these FAAL orthologues. Recently, a loss of virulence has been reported when FAAL-like domain (FLD)-containing protein, CPS1, a DIP2 orthologue, is mutated in either plant pathogenic fungi, which causes rice blast and wheat head scab, or human pathogenic fungus, that cause valley fever (*Lu et al., 2003*; *Narra et al., 2016*; *Wang et al., 2016*). Furthermore, DIP2 has been shown to be crucial for axonal branching, neurite sprouting, and dendritic spine morphogenesis in *Drosophila*, *Caenorhabditis elegans*, and mice, respectively (*Ma et al., 2019a*; *Ma et al., 2019b*; *Sah et al., 2020*; *Ma et al., 2019a*; *Nitta et al., 2017*; *Noblett et al., 2019*; *Xing et al., 2020*). Three paralogues of DIP2 in humans have been implicated as a potential risk factor for neurodevelopmental disorders like autism spectrum disorders (ASDs) and other diseases (*Egger et al., 2014*; *Gong et al., 2018*; *Iossifov et al., 2012*; *Jiao et al., 2012*; *Kong et al., 2016*; *Poelmans et al., 2009*; *Rudin et al., 2012*; *Larsson*

*et al., 2017*; *Supplementary file 1*). Despite these striking physiological consequences, our understanding of how the loss of DIP2 contributes to pathogenesis is limited by the lack of information on its cellular function.

Here, we report the previously unrecognized function for the unique FLD-containing protein, DIP2, which ensures a metabolic redirection of a defined subset of DAGs towards triacylglycerol (TAG) lipids in yeast. We provide evidence that the DIP2-mediated DAG metabolism route is conserved in *Drosophila* and mice. The mutational study suggests that the canonical enzymatic function of FLD, that is, FA activation via adenylation, is crucial for the function of DIP2 in DAG to TAG conversion. Aberrations in this metabolic flux in the absence of DIP2 trigger a homeostatic signal called unfolded protein response (UPR) pathway leading to ER stress. Interestingly, we find that DIP2 primarily localizes to mitochondria as patches while it is associated with vacuolar surface through mitochondria–vacuole contacts. Using loss and gain of function experiments, we reveal the role of DIP2-mediated DAG regulation in osmoadaptation by facilitating vacuole fusion–fission homeostasis in yeast. Overall, the work presents the discovery of a key regulator of DAG subspecies in yeast and assigns the physiological role of overlooked DAG subspecies in fundamental cellular processes, that will strengthen the emerging paradigm of lipid-mediated functional diversity.

## Results

### DIP2 is a conserved and non-canonical player in DAG metabolism across Opisthokonta

Using a phylogenetic approach, we identified that FLDs underwent two independent events of duplication (*Figure 1A*). The prokaryotic FAALs were recruited early in eukaryotes along with PKS/NRPS systems in several Protozoa, Bikonta (mainly plants and algae), and Fungi. Subsequently, a highly diverged FLD went through the first duplication to produce a tandemly fused FLD didomain in all opisthokonts (except Basidiomycota fungi). The second duplication event occurred in the vertebrates resulting in multiple paralogues of DIP2 (*Figure 1A*). Though FLDs cluster with prokaryotic and plant FAALs, they segregate as subclusters that are specific to FLD1 and FLD2 (*Figure 1B, C*). Further analyses revealed that both the FLDs retain the ancestral motifs of the ANL superfamily (*Gulick, 2009*) while gaining exclusive variations in several motifs, unique to each FLD (*Figure 1—figure supplement 1*). The divergence of Opisthokonta FLDs from prokaryotic FAALs can also be explained on the account of the low sequence identity (~17–21%) they share. For instance, Cmr2 (YOR093c), the yeast orthologue of DIP2 (hereafter referred to as ScDIP2), shares an average sequence identity of ~19% (~16–27%) with other DIP2 proteins. The tandem FLDs, FLD1 and FLD2, share an average sequence identity of ~19% and ~22% with respective FLDs from other DIP2 and ~16% for both the FLDs with the representative bacterial FAALs. Taken together, the data suggest that the duplication and divergence of FLDs have resulted in the emergence of DIP2 as a distinct gene family with possible new functions.

To begin the investigation, we generated DIP2 knock-out yeast (hereafter referred to as ΔScDIP2) using homologous recombination (*Figure 1—figure supplement 2A, B*). We hypothesized a lipidomic change in the ΔScDIP2 cells based on: (1) the presence of FLDs in ScDIP2 (*Patil et al., 2021*), which are FA-activating domains, (2) in mice, the deletion of DIP2A results in abnormal fat depositions and growth rates depending on their dietary lipid compositions (*Kinatukara et al., 2020*), and (3) analysis of the gene ontology of its genetic interactors revealed that a majority of them participate in lipid metabolism (*Figure 1—figure supplement 2C*). To address this, we radiolabelled the yeast lipids in a steady state using $^{14}$C-acetate and assessed how it fluxes through various cellular lipid pools using thin-layer chromatography (TLC) (*Figure 1D*). ΔScDIP2 cells showed a significant ~23% increase in the total DAG level compared to the wild type (*Figure 1E*) with a moderate depletion in TAG level (*Figure 1—figure supplement 2D*). However, none of the other membrane lipid classes showed any significant change (*Figure 1—figure supplement 2D*). This observation for the first time implicates the involvement of DIP2 in lipid metabolism, particularly DAG metabolism in yeast.

Subsequently, we probed other members of Opisthokonta to see if the identified role of DIP2 in DAG metabolism is conserved. We used mutants of DIP2 generated in other model organisms such as *Drosophila melanogaster* (ΔDmDIP2) (*Nitta et al., 2017*) and *Mus musculus* (ΔMmDIP2A) (*Kinatukara et al., 2020*). A comparative liquid chromatography–mass spectrometry (LC–MS) analysis of

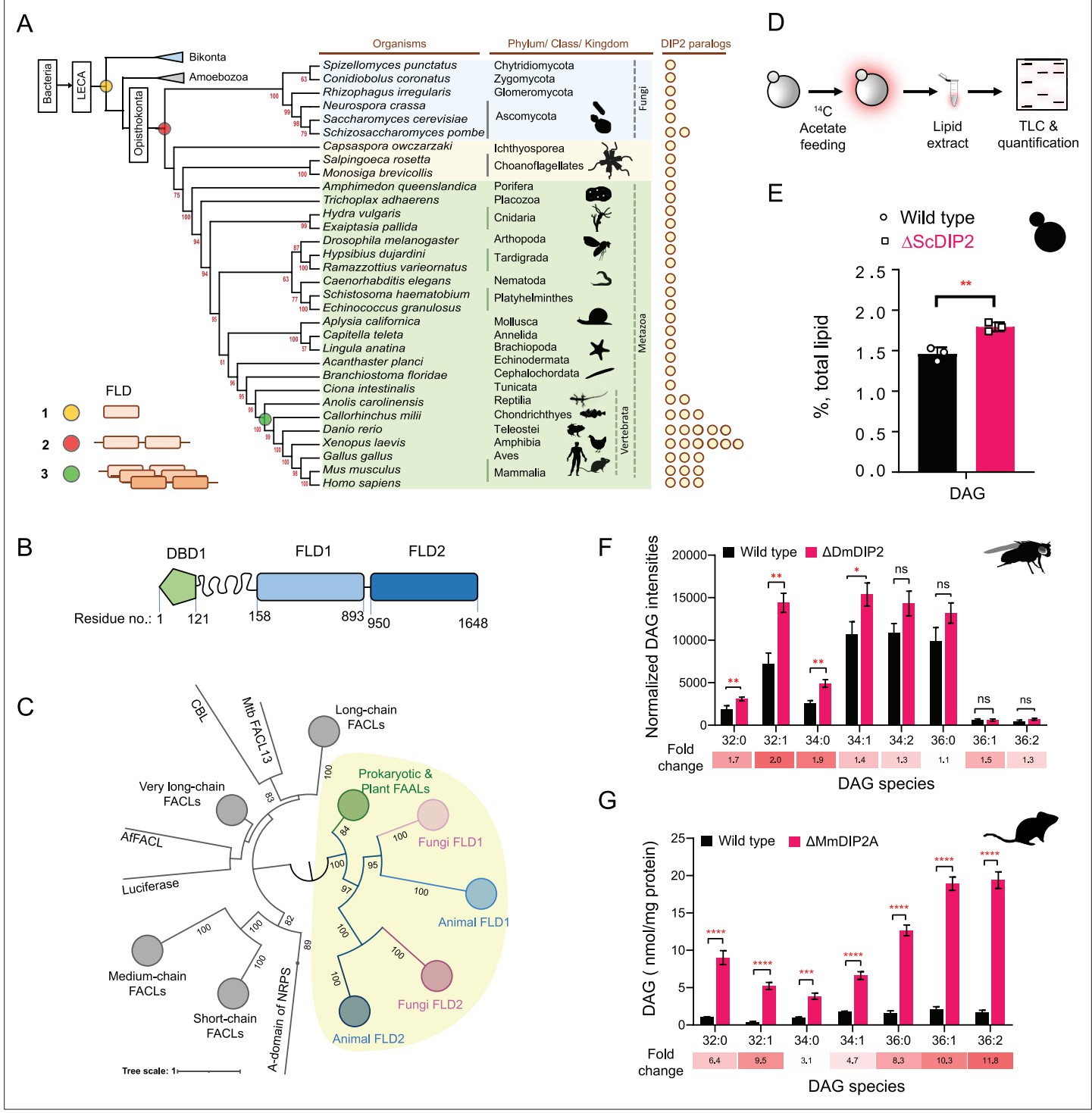

**Figure 1.** Tandem fatty acyl-AMP ligase (FAAL)-like domains (FLDs) containing protein DIP2 is a new player in the diacylglycerol (DAG) metabolism of Opisthokonta. (**A**) The distribution of FLDs in Eukaryota: three events in FLD evolution are indicated on a representative phylogenetic tree, (1) recruitment of bacteria-like standalone FLDs in eukaryotes (yellow circle), (2) domain duplication resulting in tandem FLDs in opisthokonts (red circle), and (3) whole gene duplications resulting in paralogues in vertebrates (green circle). (**B**) Domain architecture and predicted domain boundaries of yeast DIP2 homologue. Amino acid numbers are indicated for each domain boundary. DBD1, DMAP1-Binding Domain 1; FLD1, FAAL-like Domain 1; FLD2, FAAL-like Domain 2. (**C**) Phylogenetic distribution of ANL family of enzymes shows the close association of FLDs of DIP2 (FLD1 and FLD2) with prokaryotic/plant FAAL and divergence from fatty acyl-CoA ligases (FACLs) of both prokaryotic and eukaryotic origins (nodes in grey colour). FLD cluster is highlighted in a yellow background. (**D**) A schematic for the metabolic labelling of the total lipid pool by feeding the yeast with radioactive [1-$^{14}$C] acetic acid. The percentage for each lipid is calculated as a percentage of the total $^{14}$C-radiolabel incorporated in the lipidome. (**E**) The estimated DAG

*Figure 1 continued on next page*

**eLife** Research article

Biochemistry and Chemical Biology | Cell Biology

*Figure 1 continued*

levels show accumulation in a ΔScDIP2 mutant yeast in comparison to a wild type. Data are representative of three independent experiments, mean ± standard deviation (SD). \*\*p < 0.01. (**F**) Liquid chromatography–mass spectrometry (LC–MS)-based quantification of DAGs from total lipid extract from 5-day-old adult *Drosophila melanogaster* shows an accumulation of DAGs. (**G**) The accumulation of DAGs is seen in the lipidome isolated from the embryonic stem cells generated from MmDIP2A$^{-/-}$ mice in CD1 background. In **F** and **G**, the average of estimated DAGs is shown on *y*-axis and the specific lipid length (combined number of carbons in two acyl chains) of DAGs is indicated on *x*-axis. Data are represented as mean ± standard error of the mean (SEM) (*n* = 6; unpaired, two-tailed Student's *t*-test; \*p < 0.05; \*\*p < 0.01; \*\*\*p < 0.001; \*\*\*\*p < 0.0001 ns = not significant). The fold change for each DAG species is indicated below the bar graphs.

The online version of this article includes the following source data and figure supplement(s) for figure 1:

**Source data 1.** Pairwise sequence identity calculation and analysis for fatty acyl-AMP ligase (FAAL)-like domains (FLDs) from representative organisms shown in *Figure 1C*.

**Source data 2.** Quantification of total diacylglycerol (DAG) from metabolic radiolabelling experiment in yeast represented in *Figure 1E*.

**Source data 3.** Quantification of diacylglycerol (DAG) species from *Drosophila* lines shown in *Figure 1F*.

**Source data 4.** Quantification of diacylglycerol (DAG) species from mouse embryonic stem cell lines shown in *Figure 1G*.

**Figure supplement 1.** Conservation and diversification of ANL superfamily motifs in FLD1 and FLD2 of DIP2.

**Figure supplement 2.** ScDIP2 null mutant generation and metabolic radiolabelling of total lipid pool of yeast.

**Figure supplement 2—source data 1.** Uncropped labelled agarose gel images for confirmation of ScDIP2 null mutant lines shown in *Figure 1—figure supplement 2A, B*.

**Figure supplement 2—source data 2.** Full raw unedited files of *Figure 1—figure supplement 2—source data 1*.

**Figure supplement 2—source data 3.** Quantification of major lipid classes from metabolic radiolabelling experiment in yeast shown in *Figure 1—figure supplement 2D*.

whole-tissue lipid extracts from adult wild type (Canton-S) and ΔDmDIP2 *Drosophila* showed ~twofold accumulation in a subset of DAG species (*Figure 1F*). Interestingly, the absence of even a single paralogue, DIP2A in the ES cells isolated from ΔMmDIP2A mice, showed ~3- to 12-fold accumulation of DAG species compared to wild-type ES cells (*Figure 1G*). We noted a preferential accumulation of C16 acyl chain-containing subset of DAGs (C32:0, C32:1, C34:0, C34:1) in ΔDmDIP2 flies, while the accumulation is biased towards C32:1, C36:1, and C36:2 DAGs in ΔMmDIP2A ES cells (fold change indicated in *Figure 1F, G*). Taken together, these data underscore the conserved role of DIP2 in regulating the levels of DAG from fungi to mammals.

## DIP2 regulates chemically distinct DAG subspecies in yeast by facilitating their conversion to TAG

To gain further insight into the function of DIP2 in DAG metabolism, we focussed on yeast as the lipid metabolism pathways are well documented. LC–MS analysis of yeast lipidome revealed a striking ~19-fold accumulation of C36:0 and ~8-fold accumulation of C36:1 chain-length DAG subspecies in ΔScDIP2 strain (*Figure 2A* and *Figure 2—figure supplement 1A*). Genetic complementation with ScDIP2 under native or galactose-inducible (with a C-terminal GFP tag) promoters restored the levels of DAGs (*Figure 2B*). To our surprise, the abundant DAG species (representing >90% of total DAG pool), comprised of C32 and C34 chain lengths (*Casanovas et al., 2015*; *Ejsing et al., 2009*), hereafter referred to as bulk DAG pool, remained unchanged in ΔScDIP2. Interestingly, this indicates that the DAG subspecies accumulation is independent of the well-characterized bulk DAG metabolizing pathways through Dga1, Lro1, Are1, Are2 (DAG acyltransferases), and Dgk1 (DAG kinase) (*Li et al., 2020*; *Rockenfeller et al., 2018*). Therefore, we conclude that DIP2 is required for the regulation of a defined subset of the DAG pool in yeast.

DAGs are the central lipid intermediates that are channelled through *de novo* and salvage pathways to form diverse membrane and storage lipids (*Figure 2—figure supplement 1B*; *Carrasco and Mérida, 2007*). So, there is a possibility of C36:0 or C36:1 subset enrichment in other lipid classes of the ΔScDIP2 strain. However, the lipidomics profile of major phospholipids showed no alteration either in a chain-length-dependent or -independent manner (*Figure 2—figure supplement 1C–I*). Interestingly, metabolic radiolabelling showed a moderate but significant depletion (~29%) of TAG level in ΔScDIP2 (*Figure 1—figure supplement 2D*), which is a storage lipid and a major reservoir for cellular DAG pools. Lipidomics revealed a depletion of ~30–40% TAGs related to a defined subset of DAGs (C36:0 and C36:1), with the highest depletion of ~93% in the case of C18:0/36:1 TAG (*Figure 2C*

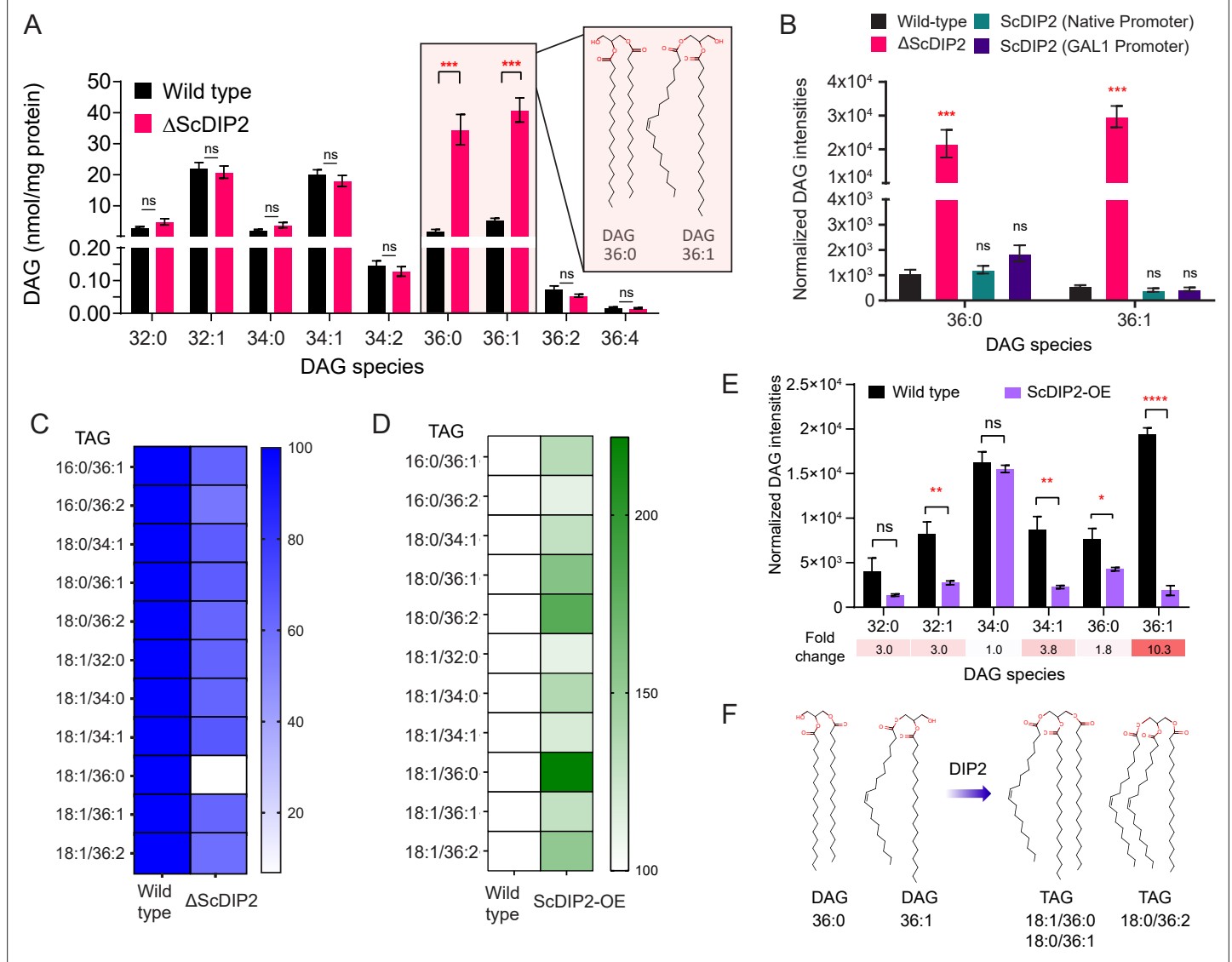

**Figure 2.** DIP2 regulates selective diacylglycerol (DAG) pools by facilitating their conversion to corresponding triacylglycerols (TAGs). (**A**) Liquid chromatography–mass spectrometry (LC–MS) analysis of the ΔScDIP2 reveals a massive accumulation of C36:0 and C36:1 DAGs. Chemical structures of indicated DAGs are shown in the inset. (**B**) Genetic complementation of the mutant with native and galactose-inducible promoter-driven ScDIP2 expression. (**C**) Deletion of ScDIP2 resulted in depletion of TAGs. The decreased level of TAGs with various chain lengths in ΔScDIP2 are shown as a linear gradient (*n* = 6) and normalized to the corresponding wild-type level (set at 100%). (**D**) Overexpression of ScDIP2 (GAL1 promoter) leads to accumulation of TAG species. The increased level of TAGs with various chain lengths in ScDIP2-OE is shown as a linear gradient (*n* = 6) and normalized to the corresponding wild-type level (set at 100%). (**E**) The DAG levels reduced significantly below the basal level upon overexpression of ScDIP2. All data are represented as mean ± standard error of the mean (SEM) (*n* ≥ 5; unpaired, two-tailed Student's *t*-test; *p < 0.05; **p < 0.01; ***p < 0.001; ****p < 0.0001; ns = not significant). The fold change for each DAG species is indicated below the bar graphs. (**F**) An illustration summarizing the role of DIP2 in facilitating the conversion of chemically distinct DAG to TAG (depicted by respective chemical structures).

The online version of this article includes the following source data and figure supplement(s) for figure 2:

**Source data 1.** Quantification of diacylglycerol (DAG) species from wild-type and ScDIP2 mutant yeast strains shown in *Figure 2A*.

**Source data 2.** Quantification of diacylglycerol (DAG) species from genetically complemented yeast strains shown in *Figure 2B*.

**Source data 3.** Quantification of triacylglycerol (TAG) species from wild-type and ScDIP2 mutant yeast strains shown in *Figure 2C*.

**Source data 4.** Quantification of triacylglycerol (TAG) species from wild-type and ScDIP2 overexpression yeast strains shown in *Figure 2D*.

**Source data 5.** Quantification of diacylglycerol (DAG) species from wild-type and ScDIP2 overexpression yeast strains shown in *Figure 2E*.

**Figure supplement 1.** Liquid chromatography–mass spectrometry (LC–MS)-based lipid profiling of yeast strains.

**Figure supplement 1—source data 1.** Quantification of diacylglycerol (DAG) species from wild-type and three biological replicates of ScDIP2 mutant

*Figure 2 continued*

yeast strains shown in *Figure 2—figure supplement 1A*.

**Figure supplement 1—source data 2.** Quantification of major phospholipid species from wild-type and ScDIP2 mutant yeast strains shown in *Figure 2—figure supplement 1C-I*.

**Figure supplement 2.** Liquid chromatography–mass spectrometry (LC–MS)-based quantification of triacylglycerol (TAG) and its lipolysis products from yeast strains.

**Figure supplement 2—source data 1.** Quantification of free fatty acid (FFA) and monoacylglycerol (MAG) species from wild-type and ScDIP2 mutant yeast strains shown in *Figure 2—figure supplement 2B, C*.

**Figure supplement 2—source data 2.** Quantification of intracellular glycerol concentration from wild-type and ScDIP2 mutant yeast strains shown in *Figure 2—figure supplement 2D*.

**Figure supplement 3.** Liquid chromatography–mass spectrometry (LC–MS)-based triacylglycerol (TAG) quantification from DIP2 knock-out mouse embryonic stem cells and *Drosophila*.

**Figure supplement 3—source data 1.** Quantification of triacylglycerol (TAG) species from mouse embryonic stem cell lines shown in *Figure 2—figure supplement 3A*.

**Figure supplement 3—source data 2.** Quantification of triacylglycerol (TAG) species from *Drosophila* lines shown in *Figure 2—figure supplement 3B*.

and *Figure 2—figure supplement 2A*). The depletion could result either from an inefficient acylation of DAGs to TAGs or increased lipolysis of those TAGs to DAGs. Under stimulated lipolysis conditions, cells are expected to attain a 3:1 ratio of free fatty acids (FFAs) to glycerol (*Schweiger et al., 2014*). The levels of FFAs, monoacylglycerols (MAGs) and glycerol were found to be unaltered in the ΔScDIP2cells (*Figure 2—figure supplement 2B–D*) when compared to the wild-type cells. It clearly suggests that increased lipolysis is unlikely to have been the cause for DAG–TAG imbalance. Taken together, the results suggest that ScDIP2 is involved in selective DAG to TAG flux, possibly by direct or indirect facilitation of an acylation reaction of DAG subspecies (*Figure 2—figure supplement 2E*). The depletion of TAG levels in ΔMmDIP2A mouse ES cell lines is in consensus with this argument (*Figure 2—figure supplement 3A*). However, it should be noted that the TAG levels of ΔDmDIP2 *Drosophila* remained comparable to wild type (*Figure 2—figure supplement 3B*), which perhaps could be attributed to a masking effect from the huge amount of TAG sourced from the fat body of *Drosophila* (*Heier and Kühnlein, 2018*).

We asked whether DIP2 has a direct effect on TAG synthesis from selective DAG subspecies. To test this, we overexpressed ScDIP2 (under GAL1 promoter), hereafter referred to as ScDIP2-OE, and checked the level of DAGs and TAGs. We found that the TAGs related to selective DAGs, C18:1/36:0 TAG increased the most (~122%), while C18:0/36:1 and C18:0/36:2 TAG increased by ~58% and ~80%, respectively in ScDIP2-OE cells (*Figure 2D* and *Figure 2—figure supplement 2A*). Concomitantly, selective DAG species, C36:1 DAG showed the highest fold change (~10-fold) in depletion, further below the basal level in ScDIP2-OE cells (*Figure 2E*). It should be noted here that a few other DAG subspecies were also depleted significantly but less drastically than the selective DAG. Such promiscuity could be a result of non-physiological levels of ScDIP2 due to overexpression. Therefore, the data suggest that the DIP2 expression is crucial to maintain the basal level of DAG subspecies and directly facilitates a DAG to TAG conversion process (*Figure 2F*).

## DIP2-mediated selective DAG regulation protects yeast from ER stress

We then sought to decipher the physiological relevance of DAG subspecies regulation by DIP2 in yeast. As ΔScDIP2 cells showed normal growth and no distinct morphological defects under standard nutrient conditions (*Figure 3—figure supplement 1A, B*), we assessed its adaptability to different stress conditions such as nutrient deficiency, osmotic stress, temperature changes, redox stress to name a few (*Figure 3—figure supplement 1C*). ΔScDIP2 cells showed significant growth impairment in the presence of tunicamycin (*Figure 3A*), which was earlier observed in a genome-wide screen (*Chen et al., 2005*). Tunicamycin, an inhibitor of *N*-linked glycosylation in the endoplasmic reticulum (ER), induces misfolded protein accumulation in ER lumen leading to proteostasis defect, commonly referred to as 'ER stress'. Genetic complementation using native promoter (*Figure 3A*) or GAL1 promoter (*Figure 3—figure supplement 2A, B*) was sufficient to rescue tunicamycin-induced ER stress. ER stress is known to activate UPR signalling, a highly conserved homeostasis signalling in all eukaryotes (*Han et al., 2010*; *Ron and Walter, 2007*). Therefore, we assessed the levels of UPR

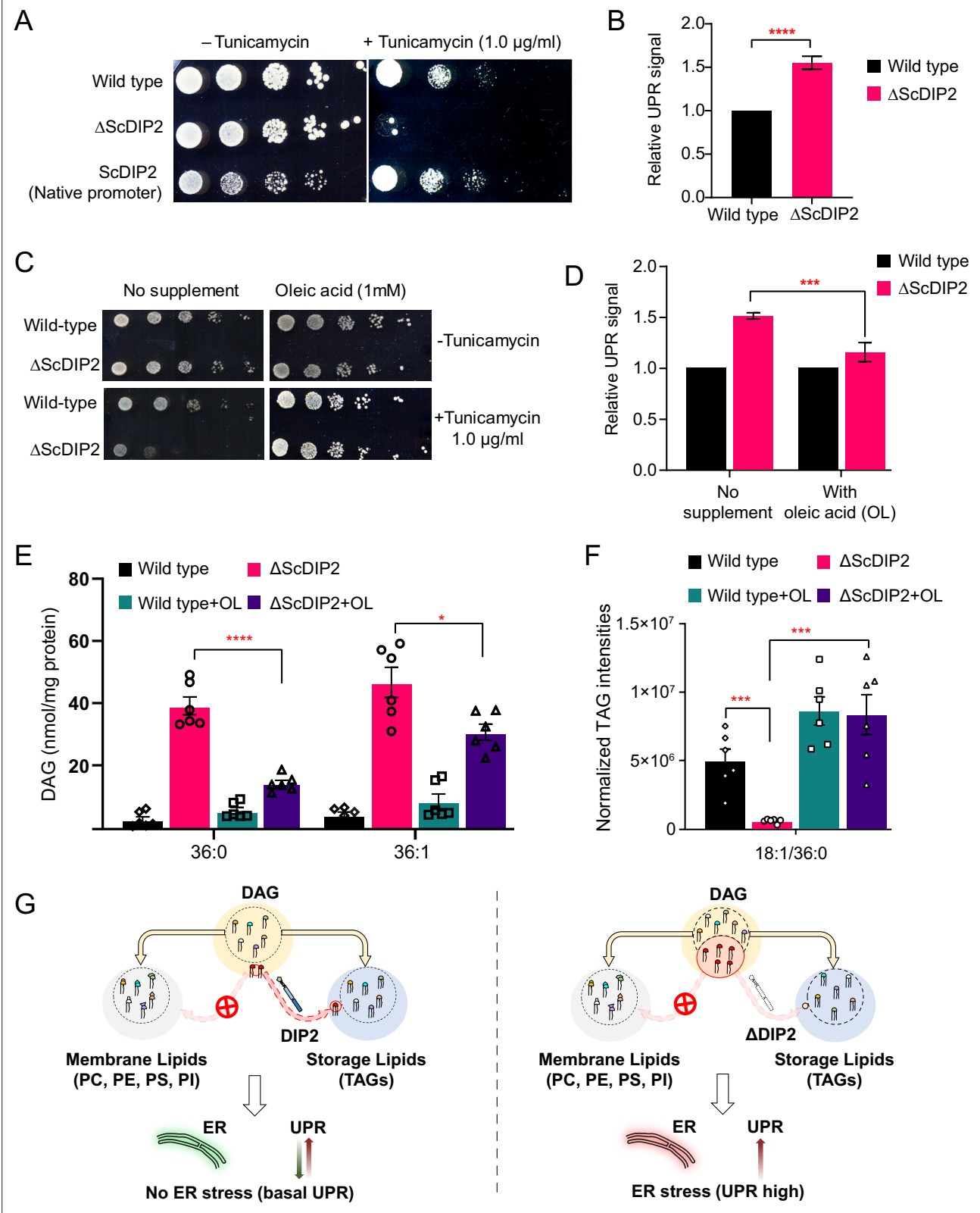

**Figure 3.** Restoration of selective diacylglycerol (DAG) level alleviates constitutive ER stress in DIP2 mutant yeast. (**A**) Serial dilution assay of yeast with indicated strains shows that ΔScDIP2 cells are sensitive to tunicamycin-induced ER stress. A representative image of the assay where expression of ScDIP2 using native promotor (genetic complementation) rescuing ER stress in ΔScDIP2 is shown. (**B**) The 4X-UPRE reporter-based quantification of unfolded protein response (UPR) signalling level. The fractional increase in RFP signal intensity (UPR signal) measured in the mutant cells from the log

*Figure 3 continued on next page*

*Figure 3 continued*

phase relative to wild-type cells in the log phase is represented. Data are represented as mean ± standard deviation (SD) (unpaired, two-tailed Student's *t*-test; n = 6; ****p > 0.0001). (**C**) Serial dilution assay with chemical supplementation of OL (1 mM) shows rescue of ER stress sensitivity in ΔScDIP2 cells. (**D**) 4X-UPRE reporter assay for wild type and ΔScDIP2 grown with 1 mM OL. The data show a reduction in UPR signal in ΔScDIP2 cells when supplemented with OL. Data are represented as mean ± SD (*n* = 4; unpaired, two-tailed Student's *t*-test; ***p < 0.001). (**E, F**) The lipidomics analysis of the oleic acid supplemented ΔScDIP2 cells shows that the accumulated C36:0 and C36:1 DAG species are significantly reduced. Corresponding triacylglycerol (TAG) species (C18:1/36:0) is also normalized to basal level upon oleic acid supplementation. Data are represented as mean ± standard error of the mean (SEM) (*n* > 5; unpaired, two-tailed Student's *t*-test; *p < 0.05; ***p < 0.001; ****p < 0.0001; ns = not significant). (**G**) A schematic representing the redirection of the selective DAG pool (red circle) to TAG by ScDIP2 helps in balancing the compositional diversity of DAG which is critical for ER homeostasis. Bulk DAG pool is utilized for membrane lipid and storage lipid (TAG) generation (shown via yellow arrow) while selective DAG pool can only be redirected to TAG (shown via red arrow).

The online version of this article includes the following source data and figure supplement(s) for figure 3:

**Source data 1.** Quantification of relative unfolded protein response (UPR) signals from yeast strains shown in *Figure 3B*.

**Source data 2.** Quantification of relative unfolded protein response (UPR) signals from yeast strains supplemented with oleic acid shown in *Figure 3D*.

**Source data 3.** Quantification of diacylglycerol (DAG) species from yeast strains supplemented with oleic acid shown in *Figure 3E*.

**Source data 4.** Quantification of triacylglycerol (TAG) species from yeast strains supplemented with oleic acid shown in *Figure 3F*.

**Figure supplement 1.** Growth phenotype and stress responses by ΔScDIP2 cells.

**Figure supplement 1—source data 1.** Growth curve assay for wild-type and ScDIP2 mutant yeast strains shown in *Figure 3—figure supplement 1A*.

**Figure supplement 1—source data 2.** Cell viability assay for wild-type and ScDIP2 mutant yeast strains shown in *Figure 3—figure supplement 1B*.

**Figure supplement 1—source data 3.** Validation of UPRE reporter assay shown in *Figure 3—figure supplement 1E, F*.

**Figure supplement 2.** Genetic complementation and chemical supplementation assay.

**Figure supplement 2—source data 1.** Uncropped labelled western blot images showing the expression of full-length ScDIP2 protein under GAL1 promoter in yeast shown in *Figure 3—figure supplement 2B*.

**Figure supplement 2—source data 2.** Full raw unedited files of *Figure 3—figure supplement 2—source data 1*.

**Figure supplement 2—source data 3.** Quantification of diacylglycerol (DAG) species from yeast strains supplemented with choline shown in *Figure 3—figure supplement 2E*.

**Figure supplement 2—source data 4.** Quantification of triacylglycerol (TAG) species from yeast strains supplemented with oleic acid shown in *Figure 3—figure supplement 2F*.

**Figure supplement 2—source data 5.** Quantification of fold change in triacylglycerol (TAG) species upon oleic acid supplementation in wild-type and ScDIP2 null mutant yeast shown in *Figure 3—figure supplement 2G*.

**Figure supplement 3.** ER stress sensitivity assay with null mutants of bulk diacylglycerol (DAG) metabolizing enzymes.

activation, using fluorescence-activated cell sorting (FACS)-based assay (*Jonikas et al., 2009*; *Merksamer et al., 2008*; *Surma et al., 2013*). The intensity of the fluorescence signal from the reporter is a direct measure of the levels of UPR activation when the cells are subjected to ER stress and the assay was validated by titrating with tunicamycin (*Figure 3—figure supplement 1D, E*). ΔScDIP2 cells exhibited a ~1.6-fold elevated UPR stress than basal levels found in wild type (*Figure 3B*) and the addition of tunicamycin only exacerbated the already active ER stress in the mutant (*Figure 3—figure supplement 1F*). It implies that the mutant suffers from ER stress even in the absence of stress-inducing agents.

Since the link between UPR and lipid metabolism is well established (*Volmer and Ron, 2015*; *Volmer et al., 2013*), aberrant DAG metabolism in ΔScDIP2 cells could be ascribed for the ER proteostasis defect. Therefore, we hypothesized that by metabolically fluxing these DAGs into other lipids, we may simply be able to alleviate the ER stress in ΔScDIP2 cells. It is known that specific lipid precursors such as choline, ethanolamine, and FAs can facilitate the utilization of cellular DAG pool to produce phosphatidylcholine (PC), phosphatidylethanolamine (PE), and TAGs, respectively (*Boumann et al., 2004*; *Henneberry et al., 2001*; *Montell et al., 2001*). Therefore, we performed a 'chemical complementation' screening to rescue the ER stress in ΔScDIP2 cells by supplementing with lipid precursors. We found that except for oleic acid (OL; C18:1 FFA), none of the other precursors rescued the ER stress phenotype (*Figure 3C* and *Figure 3—figure supplement 2C*). The supplementation of ΔScDIP2 mutants with a different unsaturated FA (e.g. palmitoleic acid; C16:1 FFA) or saturated FA (e.g.: palmitic acid; C16:0 FFA, stearic acid; C18:0 FFA) did not relieve ER stress (*Figure 3—figure supplement 2D*). Furthermore, OL supplementation was enough to reduce the constitutive

UPR upregulation in ΔScDIP2 (*Figure 3D*). In agreement with those observations, the LC–MS analysis revealed the restoration of the accumulated subset of DAGs back to basal levels (*Figure 3E*). A concomitant normalization in related TAG subspecies (*Figure 3F* and *Figure 3—figure supplement 2F*) was also observed upon OL supplementation along with an overall ~three- to sevenfold increase in bulk TAG production (*Figure 3—figure supplement 2G*). All the other precursors showed little or no effect on the levels of an accumulated subset of DAGs (*Figure 3—figure supplement 2E*), which also correlated with their inability to suppress tunicamycin sensitivity. It is interesting to note that bulk DAG accumulation is possibly less toxic to ER as the null mutants of canonical DAG-utilizing enzymes such as Dga1 and Lro1 or Dgk1 are not sensitive to ER stress (*Figure 3—figure supplement 3*). This indicates the atypical role of selective DAG subspecies as their accumulation results in ER stress, while ScDIP2 alleviates the ER stress by facilitating the buffering of such DAGs from the membrane pool to the storage pool (*Figure 3G*).

Since OL supplementation was found to restore DAG level, we wondered if any physiological condition(s) can mimic this process. Under nutrient limitation conditions such as stationary phase, yeast is known to massively upregulate endogenous OL production and subsequent TAG synthesis, facilitated by majorly Dga1 and Lro1, as a metabolic strategy to store energy in the form of neutral lipids (*Casanovas et al., 2015*; *Jacquier et al., 2011*; *Joshi et al., 2018*; *Markgraf et al., 2014*). To test this, the DAG levels in the early-log, mid-log, and late stationary phases of wild type were compared with the levels in ΔScDIP2 cells. The bulk DAG levels across the various stages of growth phases remain constant in the wild type, which was also observed in earlier reports (*Casanovas et al., 2015*). Strikingly, the ΔScDIP2 cells showed the accumulation of a selective subset of DAGs only during the early- and mid-log phase and these levels are restored during the stationary phase (*Figure 4A*). Thus, the toxic accumulation of selective DAGs is diluted by the bulk conversion of DAGs to TAGs during the stationary phase, which also suppress the activated UPR (*Figure 4B*). It is, therefore, evident that the role of ScDIP2 in regulating a defined subset of DAGs to TAG flux must be a growth phase-dependent phenomenon. So, we tracked the expression of ScDIP2 along the growth phase using a ScDIP2-GFP knock-in line (C-terminal GFP). The expression of ScDIP2-GFP appeared as an intracellular tubular and punctate-like structure only during the early- to mid-log phase and rapidly diminished in the stationary phase (*Figure 4C*). Taken together, these observations reveal an interesting parallel between canonical and ScDIP2-dependent regulation of DAG subspecies required to protect ER function in the logarithmic growth phase (*Figure 4D*).

## DIP2-regulated selective DAG species are required for vacuole-fusion-mediated osmoadaptation

Since DAGs are metabolized at multiple organellar sites (*Baron and Malhotra, 2002*; *Cowell et al., 2009*; *Ganesan et al., 2019*; *Starr and Fratti, 2019*; *Yang and Kazanietz, 2003*), the subcellular location of ScDIP2 may indicate the physiological niche of selective DAGs. A subcellular fractionation assay showed the association of ScDIP2 with the total membrane fraction isolated from log-phase cells (*Figure 5—figure supplement 1E*). Using fluorescence microscopy, ScDIP2-GFP signal was found to be primarily colocalized with the mitochondrial marker, MitoTracker (*Figure 5—figure supplement 1A, B*). This observation was also supported by previous reports on mitochondrial colocalization of mouse DIP2A in a mammalian cell line and its role in superoxide dismutase-mediated antioxidative response (*Bai et al., 2021*; *Ma et al., 2019a*). Interestingly, we also observed a partial association of ScDIP2-GFP puncta with vacuolar surface stained with FM4-64 dye (*Figure 5—figure supplement 1C, D*; see also *Figure 5—figure supplement 1J–M*). A quantification of colocalization revealed that 30–40% of ScDIP2-GFP puncta (fluorescence signal) overlapped with vacuolar marker and 70–80% signal overlapped with mitochondrial marker (*Figure 5—figure supplement 1F, G*). Subsequently, a three-colour colocalization study of ScDIP2-GFP (green), mitochondria (4′,6-diamidino-2-phenylindole (DAPI); blue) (*Dellinger and Gèze, 2001*; *Higuchi-Sanabria et al., 2016*; *Zarin et al., 2021*), and vacuoles (FM4-64; red) showed that a fraction of ScDIP2-GFP puncta (~30%) was associated with both the vacuolar and mitochondrial marker signal (*Figure 5A, B*; *Figure 5—figure supplement 11H, 1I*). While the major fraction of ScDIP2-GFP puncta localizes to mitochondria, all the vacuole-associated ScDIP2 puncta overlap with mitochondrial marker (DAPI), suggesting that the vacuolar association of ScDIP2 is at the mitochondrial–vacuolar contact site. The localization of ScDIP2 at mitochondria–vacuole contact site is further supported by a previous study

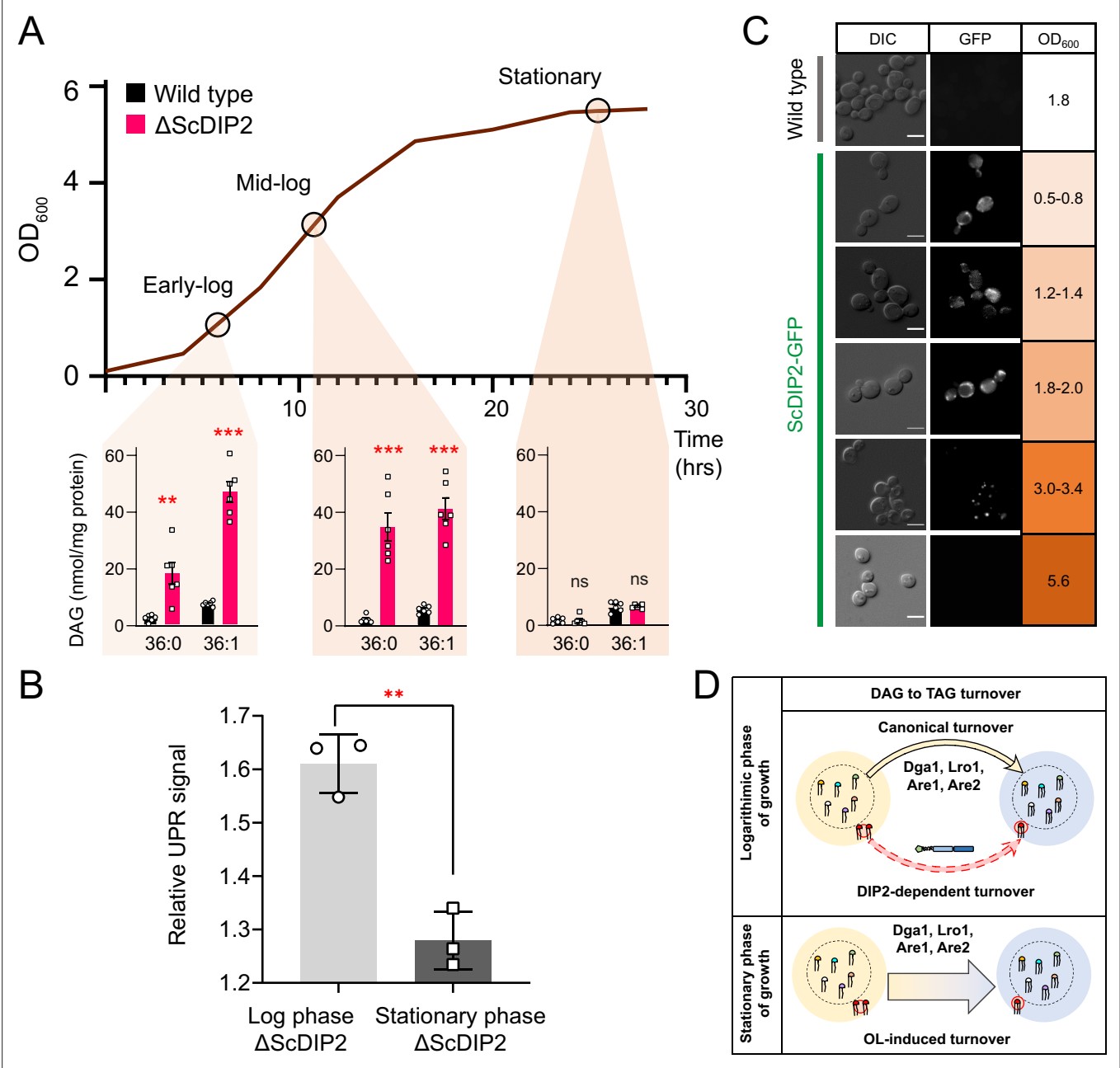

**Figure 4.** Growth phase-dependent regulation of selective diacylglycerols (DAGs) by DIP2 ensures ER homeostasis in yeast. (**A**) Tracking the level of selective DAGs in ΔScDIP2 cells across the growth phases: early-log, mid-log, and stationary phase. Data are represented as mean ± standard error of the mean (SEM) ($n > 5$; unpaired, two-tailed Student's $t$-test; **$p < 0.01$; ***$p < 0.001$; ns = not significant). (**B**) The 4X-UPRE reporter assay shows a reduced level of unfolded protein response (UPR) activation when ΔScDIP2 cells are grown till the stationary phase. Data represent fold change in ΔScDIP2 relative to respective wild-type control (mean ± standard deviation [SD]; $n = 3$; unpaired, two-tailed Student's $t$-test; **$p < 0.01$). (**C**) Growth-phase-dependent expression of ScDIP2 is shown via the fluorescence signal of GFP tagged with C-terminal ScDIP2 in its genomic locus. Scale bar = 5 µm. (**D**) A schematic showing DIP2-dependent selective DAG to triacylglycerol (TAG) turnover is operative in the log phase while a surge in OL-induced TAG biosynthesis during the stationary phase restores the overall DAG level.

The online version of this article includes the following source data for figure 4:

**Source data 1.** Quantification of diacylglycerol (DAG) species from wild-type and ScDIP2 mutant yeast strains across growth phases shown in *Figure 4A*.

**Source data 2.** Quantification of relative unfolded protein response (UPR) signals from yeast strains at stationary phase shown in *Figure 4B*.

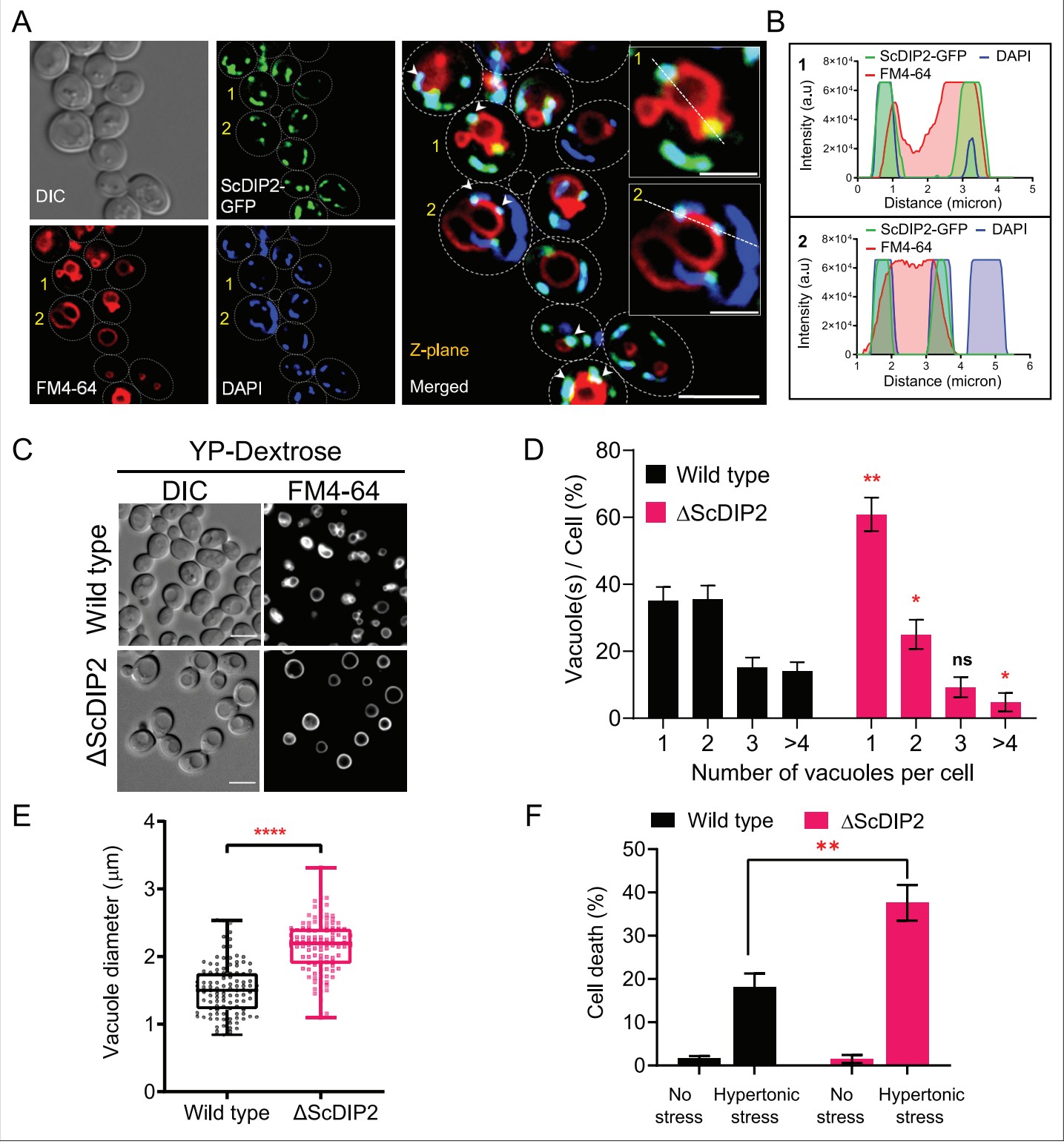

**Figure 5.** Loss of DIP2 results in aberrant vacuolar fusion leading to failure in osmoadaptation in yeast. (**A**) Live cell imaging of ScDIP2-GFP knock-in yeast strain for subcellular localization study. ScDIP2-GFP cells (at logarithmic growth phase) are stained with vacuole membrane marker, FM4-64 (red) dye and DAPI to visualize mitochondrial DNA. The images were obtained using a widefield epifluorescence microscope. The enlarged images of representative cells (marked with 1 and 2) are shown in inset. Single plane image is displayed, and white arrowhead denotes the colocalization sites. Cell outlines were drawn over DIC images. Scale bar represents 5 μm for merged image and 2 μm for inset images. (**B**) Line scan of signal intensity along the white lines shown in inset images of (**A**), marked with 1 and 2. (**C**) Representative images of vacuole morphology of the indicated strains grown till

*Figure 5 continued on next page*

*Figure 5 continued*

log phase in YP-dextrose media. Cells were stained with FM4-64. Scale bar = 5 µm. (**D**) Quantitation of vacuoles observed per cell of indicated strains imaged in (**C**). Data were represented as mean ± standard deviation (SD) ($n$ = 3; >100 cells per strain; unpaired, two-tailed Student's $t$-test; *$p$ < 0.05; **$p$ < 0.01; ns, not significant). (**E**) Vacuole diameter has been measured by line scan analysis of FM4-64-stained vacuoles from yeast cells at the budding stage. Each dot represents a single vacuole per cell (>100 cells per strain) and box and whisker plot representing diameters (in µm) of the largest vacuole of indicated strains. Data are representative of at least three independent experiments (unpaired, two-tailed Student's $t$-test; ****$p$ < 0.0001). (**F**) ΔScDIP2 and wild-type cells are subjected to hypertonic stress and then stained with trypan blue for assessing cell viability. Data are represented as mean ± SD ($n$ > 3; unpaired, two-tailed Student's $t$-test; *$p$ < 0.05; **$p$ < 0.01; ****$p$ < 0.0001; ns = not significant).

The online version of this article includes the following source data and figure supplement(s) for figure 5:

**Source data 1.** Percentage of yeast cell harbouring different numbers of vacuoles in logarithmic phase of growth shown in *Figure 5D*.

**Source data 2.** Quantification of vacuole diameter from wild-type and ScDIP2 mutant yeast strains shown in *Figure 5E*.

**Source data 3.** Percentage of yeast cell death upon hypertonic stress shown in *Figure 5F*.

**Figure supplement 1.** Subcellular localization study of ScDIP2.

**Figure supplement 1—source data 1.** Uncropped labelled western blot images showing subcellular association of ScDIP2 protein shown in *Figure 5—figure supplement 1E*.

**Figure supplement 1—source data 2.** Full raw unedited files of *Figure 5—figure supplement 1—source data 1*.

**Figure supplement 1—source data 3.** Quantification of colocalization between ScDIP2-GFP puncta and vacuole and mitochondria marker signal shown in *Figure 5—figure supplement 1F, G*.

showing its interaction with vacuole–mitochondria contact site (vCLAMP) protein, Vps39 (*Elbaz-Alon et al., 2014*).

As previous studies highlighted mitochondrial localization, we focussed our study on the unexplored impact of the association of ScDIP2 with vacuole through mitochondria–vacuole contact sites. Since DAGs are known to affect vacuole function, we hypothesized that loss of ScDIP2 would affect vacuole function. Therefore, we examined the vacuole morphology in the log phase as it serves as an index of vacuole health. We found that about 60% of ΔScDIP2 cells contain a greater number of single, rounded, and enlarged vacuoles, which is roughly twice that of the wild type (*Figure 5C, D*). A comparative analysis of the size distribution of vacuoles in the log phase of growth showed that larger sized vacuoles are more frequently observed in ΔScDIP2 mutant than the wild type (*Figure 5E*). Thus, the increase of single and large vacuoles in ΔScDIP2 cells is presumably due to aberrant vacuole membrane fusion driven by accumulated DAGs, a known membrane fusogen (*Starr and Fratti, 2019*). The optimum vacuole fusion is critical for cells' adaptation to osmotic stress, a common environmental stress experience by fungi (*Veses et al., 2008*), To check the functional consequence of such compromised vacuolar homeostasis, ΔScDIP2 cells were subjected to hyper-osmotic stress using high-salt growth media. An incubation in hyper-osmotic conditions resulted in two- to threefold higher cell death in ΔScDIP2, while the cell viability was identical for both the strains in the absence of stress (*Figure 5F*). A similar observation was also made during the screening of ΔScDIP2 using serial dilution assay (*Figure 3—figure supplement 1C*).

Next, we investigated if DAG depletion upon ScDIP2 overexpression can show converse effects on vacuole morphology. The vacuoles in the ScDIP2-OE strain were found to be highly fragmented; single vacuole containing cells are about three times lower while the multivacuolar (>4) cells were three times more than that in the wild type (*Figure 6A, B*). To confirm whether this fragmented vacuolar morphology is due to the impairment of vacuole fusion, we employed a vacuole fusion–fission assay utilizing the osmotic response of yeast (*Figure 6C*). The growth in hypertonic media induces vacuole fission in yeast, resulting in highly fragmented vacuoles in both wild-type and ScDIP2-OE cells (*Figure 6D, E*). Exposure of such cells with fragmented vacuoles to hypotonic conditions induces rapid vacuole fusion. Vacuoles in ScDIP2-OE cells are found to be highly fragmented and multilobar even after the fusion induction using hypotonic media (*Figure 6F*), suggesting a significantly reduced vacuolar fusion process (~30% reduced fusion compared to wild type) (*Figure 6G*). Since rapid vacuole fusion is required during hypo-osmotic stress adaptation by yeast cells, we hypothesized that ScDIP2-OE cells will show reduced fitness in hypotonic conditions. Indeed, we found that acute hypotonic stress caused severe cell death in ScDIP2-OE while ΔScDIP2 cells showed significant tolerance to the stress compared to the wildtype (*Figure 6H*). Together, the data suggest that the selective DAG

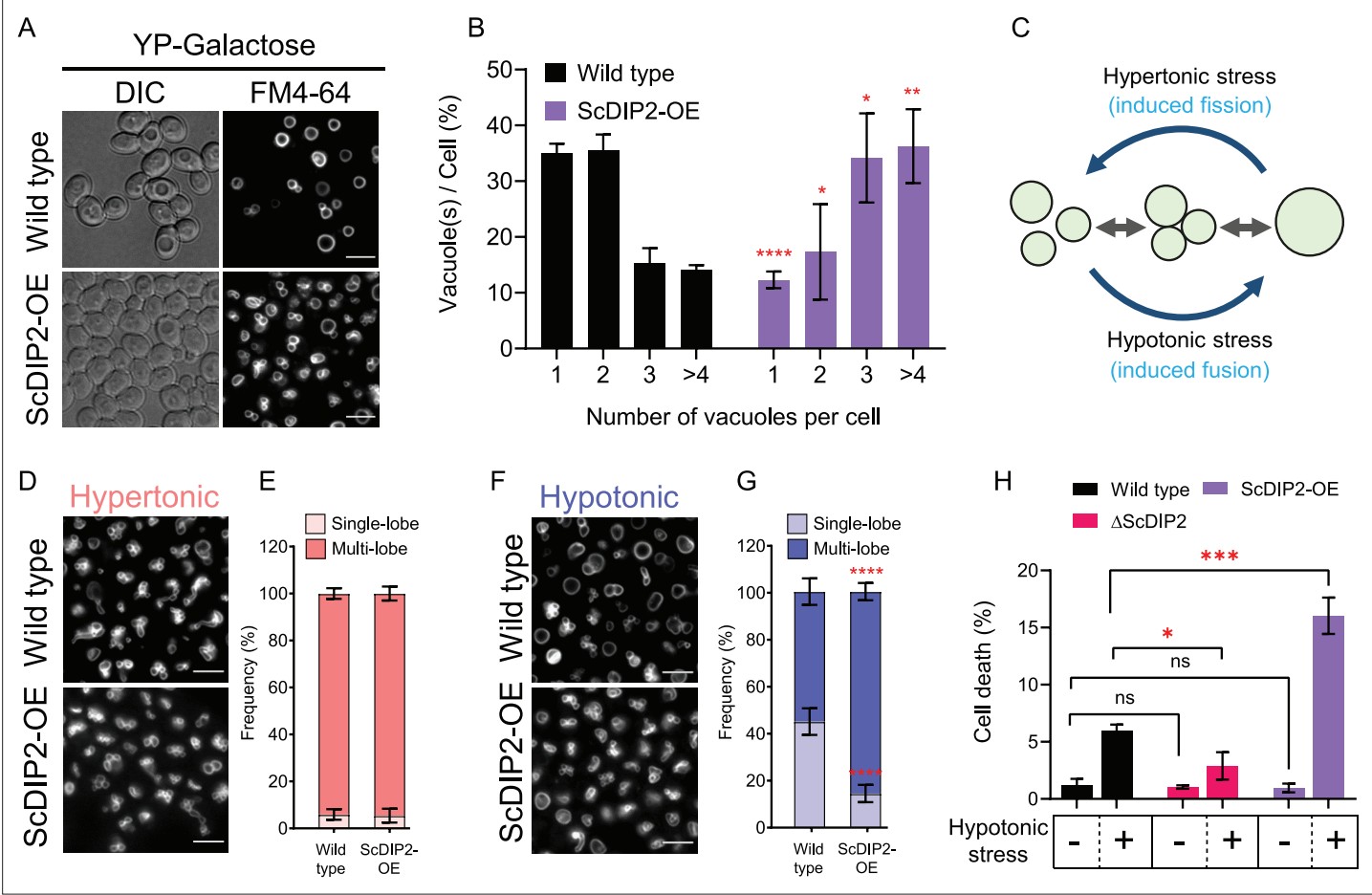

**Figure 6.** Depletion of diacylglycerol (DAG) upon DIP2 overexpression destabilizes vacuolar fusion–fission homeostasis in yeast. (**A**) Representative images of vacuole morphology in indicated strains grown till log phase in YP-galactose media. Vacuoles are stained with FM4-64. Scale bar = 5 μm. (**B**) Quantitation of vacuoles observed per cell of indicated strains imaged in (**A**). Data are represented as mean ± standard deviation (SD) ($n = 3$; >100 cells per strain; unpaired, two-tailed Student's $t$-test; *$p < 0.05$; **$p < 0.01$; ****$p < 0.0001$; ns = not significant). (**C**) A schematic depicting the vacuole fusion–fission assay utilizing the osmotic response of yeast to different concentrations of NaCl. (**D, F**) Representative images of FM4-64-stained vacuoles from indicated strains after treating with hypertonic (YPD + 0.4 M NaCl) condition for 10 min, followed by hypotonic (water) condition for 10 min. (**E, G**) Quantification of vacuoles in hypertonic conditions (shades of pink) and hypotonic conditions (shades of blue). Frequency percentage of single-lobe (fused) and multi-lobe (fragmented) vacuole per cell from indicated strains imaged in D and F. Cells with three or more vacuoles were combined as 'multi-lobe vacuole' set ($n = 6$, >100 cells per strain; unpaired, two-tailed Student's $t$-test; ****$p < 0.0001$). (**H**) Indicated strains are subjected to hypotonic shock by incubating in water for 10–15 min and then stained with trypan blue for cell viability. Data are represented as mean ± SD ($n > 3$: two-tailed Student's $t$-test; *$p < 0.05$; **$p < 0.01$; ***$p < 0.001$; ****$p < 0.0001$; ns = not significant).

The online version of this article includes the following source data for figure 6:

**Source data 1.** Percentage of yeast cell harbouring different numbers of vacuoles in logarithmic phase of growth shown in *Figure 6B*.

**Source data 2.** Quantification of induced vacuole fusion in yeast strains shown in *Figure 6E, G*.

**Source data 3.** Percentage of yeast cell death upon hypotonic stress shown in *Figure 6H*.

pool is essential for the vacuole fusion process and ScDIP2-mediated regulation optimizes the process to ensure osmoadaptation in yeast.

## Catalytic activity of tandem FLD didomain is critical for DAG regulation function

Next, we asked whether the FLDs of DIP2 are required for the DAG regulation. Bioinformatics analysis on the emergence of the unique three-domain architecture of DIP2 in opisthokonts revealed that the N-terminal DMAP1-binding domain (DBD1) is absent in several ancient members of this group (viz. Nucleariids, Ichthyosporea, Choanoflagellates, and some fungi species). However, the tandem

FLD-didomain (FLD1–FLD2) architecture is conserved throughout all opisthokonts (*Figure 7A*). For genetic complementation, we generated multiple GAL1-based domain-truncated constructs of ScDIP2 and their expression was validated by fluorescence microscopy and western blot experiments (*Figure 7—figure supplement 1A, B*). The genetic complementation assay with multiple domain-truncated constructs revealed that tandem FLD didomain is minimally required to prevent ER stress, while the DBD1 is dispensable (*Figure 7B*). We also performed lipidomics of ΔScDIP2 yeast complemented with different domain-truncated versions of ScDIP2. We found that the FLD didomain restored the levels of the subset of DAGs characteristic to yeast (*Figure 7C*), accompanied by the generation of related TAGs (*Figure 7D*). Such a restoration of the DAG level is not evident when either of the FLDs is missing (*Figure 7—figure supplement 1C*). These results suggest that the recruitment and duplication of FLDs was an innovation in early Opisthokonta evolution to tackle a specific fraction of the DAGs pool, which if accumulated may become toxic.

To delineate if the FLDs have retained the enzymatic activity of a typical ANL superfamily module in DAG regulation, we performed sequence and mutational analysis of FLDs. A typical ANL superfamily member, including FLDs, should activate their respective substrates by the formation of an acyl-adenylate intermediate. All these homologues show highly conserved residues responsible for the acyl-adenylate formation, referred to as A5 and A7 motifs (*Gulick, 2009*). Structure-based sequence alignments, using models from AlphaFold Protein Structure Database (*Jumper et al., 2021*; *Varadi et al., 2022*), suggested that the A5 and A7 motifs are conserved in the FLD1 domains of all opisthokonts (*Figure 7E*) and strikingly, FLD2 showed degeneration of both the motifs. This is also supported by the greater divergence of FLD2 domains from the prokaryotic/plant FAALs than FLD1 domains, as inferred from the branch lengths of FLD clades in phylogenetic tree (*Figure 1C*). Mutations in catalytic residues of adenylation motifs of FLD1 (*Figure 7E* and *Figure 7—figure supplement 2A*) led to the loss of function of ScDIP2. These adenylation mutants failed to rescue the ΔScDIP2 from tunicamycin-induced ER stress (*Figure 7—figure supplement 2D*) and also showed an inability to regulate the levels of accumulated DAGs by converting them to related TAGs (*Figure 7F, G*). Overexpression of these mutants neither resulted in fragmented vacuole morphology nor an increased cell death in hypotonic stress condition as seen with wild-type ScDIP2 overexpression (*Figure 7—figure supplement 2F, G*). The expression and the localization pattern of FLD1 adenylation mutants were identical to the wild-type ScDIP2 (*Figure 7—figure supplement 2B, C*). The mutation of predicted residues of the degenerate motifs in FLD2 did not lead to the loss of function (*Figure 7—figure supplement 2B, C, E*). Taken together, the adenylation activity of FLD1 of DIP2 is required for facilitating the conversion of selective DAGs to TAGs. Also, it can be suggested that FLD2 has not retained adenylation activity and has diverged for a yet to be identified function.

## Discussion

'Why do cells possess a highly diverse lipidome?' is one of the most perplexing questions in biology. Diversity amongst various classes of lipids arising from the variability in FA compositions such as chain length, degree of unsaturation, is turning out to be crucial for biological functions (*Atilla-Gokcumen et al., 2014*; *Khandelwal et al., 2021*; *Raghu, 2020*; *Shin et al., 2020*). Earlier *in vivo* studies have pointed to the exclusive role of selective DAG subspecies in determining the fate of different physiological processes including signal transduction, cell polarity, membrane trafficking, ligand binding to membrane receptors, and modulation of membrane properties (*Lee et al., 1991*; *Marignani et al., 1996*; *Schuhmacher et al., 2020*; *Ware et al., 2020*). However, the molecular players and pathways in selective regulation are largely unknown. Interestingly, studies in mammals have revealed the presence of a unique isoform of diacylglycerol kinase, namely DGKε, that channelizes specific DAG species, suggesting the evolution of selective metabolism routes in eukaryotes (*Milne et al., 2008*). However, the known DAG metabolizing enzymes of yeast, DAG acyltransferase (Dga1, Lro1, Are1, and Are2) and DAG kinases (Dgk1), are non-specific in nature and thus facilitate a generic regulation through bulk conversion of DAGs (*Li et al., 2020*; *Rockenfeller et al., 2018*). Therefore, a thorough investigation is needed to identify how ~2 billion years of eukaryotic evolution has shaped the lipid repertoire through selective regulation mechanisms. In this context, our study demonstrates the cellular function of a conserved protein family, DIP2 in regulating a subset of DAG pool (C36:0 and C36:1 DAGs) of yeast in a growth phase-dependent manner. In yeast, the loss of DIP2 leads to the accumulation of these DAG subspecies and a concomitant depletion of chain-length-related TAG species. These DAG

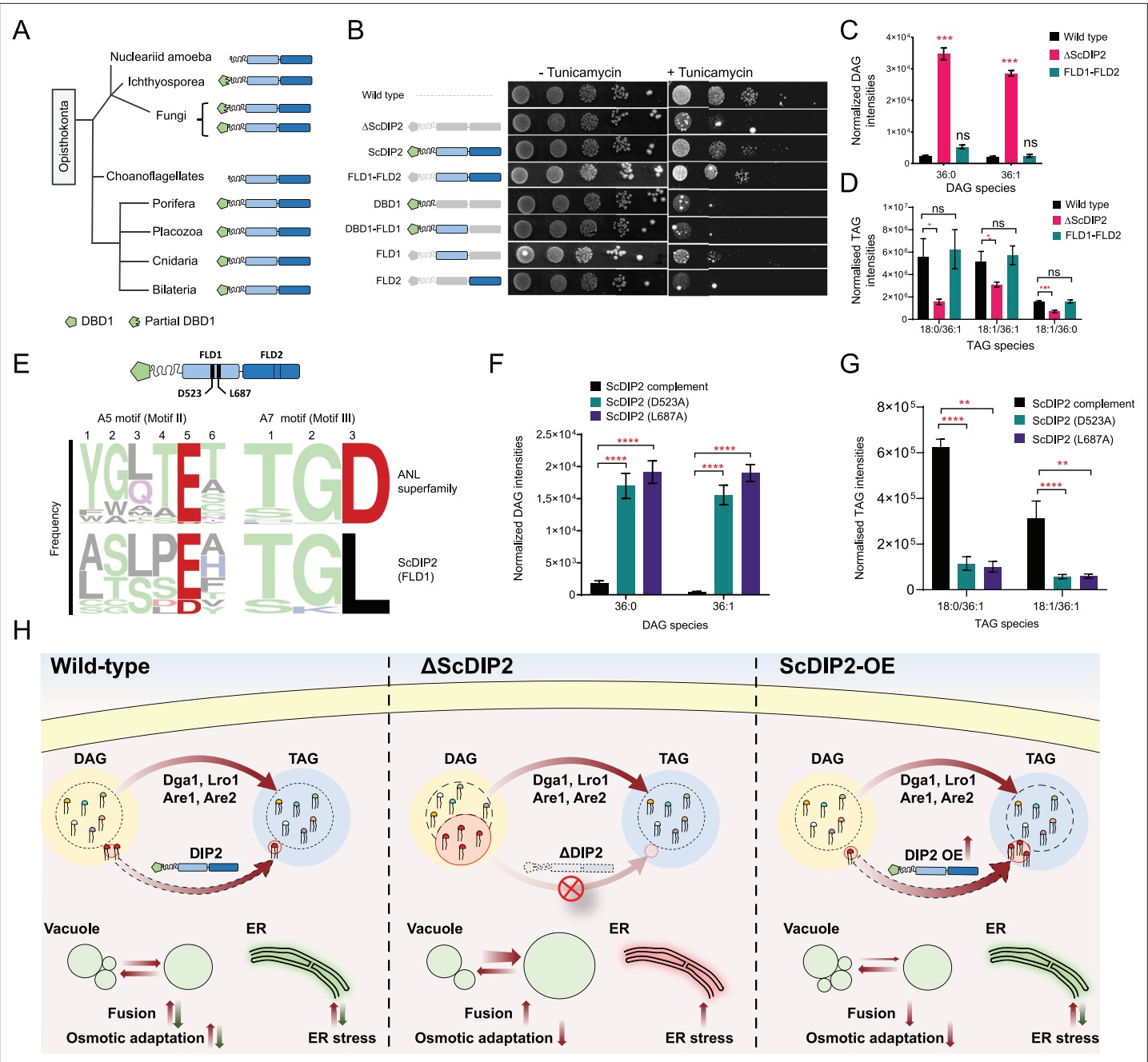

**Figure 7.** Catalytic activity along with tandem FLD-didomain organization is critical for the function of DIP2. (**A**) A graphical representation of the phylogenetic distribution of DIP2 across Opisthokonta showing conservation of FLD1–FLD2 didomain. (**B**) Tunicamycin-mediated ER stress rescue assay by complementing ΔScDIP2 with different domain-truncated versions of ScDIP2. Deleted domains are shown in grey colour. (**C, D**) Increased diacylglycerols (DAGs) in ΔScDIP2 are restored to wild-type level upon complementation with FLD1–FLD2 didomain, along with an increase in related triacylglycerol (TAG) levels. (**E**) Graphical representation to show the crucial residues of adenylation motifs of ScDIP2 FLD1 and sequence logo images depicting the conservation of adenylation motifs of ANL superfamily in DIP2. (**F**) Adenylation mutants of ScDIP2 are not able to restore accumulated DAGs to basal level due to failure in fatty-acyl activation. (**G**) Selective TAG level is not restored to wild-type level upon expression of adenylation mutants of ScDIP2. Data are represented as mean ± standard error of the mean (SEM) ($n > 5$; unpaired, two-tailed Student's $t$-test; *$p < 0.05$; **$p < 0.01$; ***$p < 0.001$; ****$p < 0.0001$; ns = not significant). (**H**) A model depicting the role of DIP2-mediated selective DAG regulation for cellular homeostasis and adaptation. The DAG subspecies, representing a small fraction (shown in red head colours) of the total DAG pool, are regulated without altering the levels of bulk DAGs (multiple head colours) by redirecting them to TAGs (shown in red) which is facilitated by DIP2 in yeast. Such a regulation of minor DAG subspecies is essential for optimal vacuole (shown as green circle) fusion–fission dynamics leading to efficient osmotic adaptation and maintenance of ER homeostasis. In ΔScDIP2, the accumulation of these minor DAG subspecies results in ER stress, increased vacuole fusion and

*Figure 7 continued on next page*

*Figure 7 continued*

reduced osmotic adaptation capability. On the contrary, in ScDIP2-OE, these DAGs are depleted below the basal level, resulting in restored ER stress but reduced vacuole fusion along with the defect in osmotic adaptation.

The online version of this article includes the following source data and figure supplement(s) for figure 7:

**Source data 1.** Quantification of diacylglycerol (DAG) species from FLD-didomain complemented yeast strains shown in *Figure 7C*.

**Source data 2.** Quantification of triacylglycerol (TAG) species from FLD-didomain complemented yeast strains shown in *Figure 7D*.

**Source data 3.** Quantification of diacylglycerol (DAG) species from yeast strains complemented with adenylation mutants of ScDIP2 shown in *Figure 7F*.

**Source data 4.** Quantification of triacylglycerol (TAG) species from yeast strains complemented with adenylation mutants of ScDIP2 shown in *Figure 7G*.

**Figure supplement 1.** Expression validation and lipidomics of ScDIP2 domain-truncated constructs.

**Figure supplement 1—source data 1.** Uncropped labelled image showing expression of ScDIP2 protein and its different domain-truncated versions shown in *Figure 7—figure supplement 1B*.

**Figure supplement 1—source data 2.** Full raw unedited files of *Figure 7—figure supplement 1—source data 1*.

**Figure supplement 1—source data 3.** Quantification of diacylglycerol (DAG) species from FLD1 and FLD2 complemented yeast strains shown in *Figure 7—figure supplement 1C*.

**Figure supplement 2.** Probing the catalytic role of ScDIP2 via mutational analysis of FLDs.

**Figure supplement 2—source data 1.** Uncropped labelled image showing expression of ScDIP2 protein and its adenylation mutant versions shown in *Figure 7—figure supplement 2C*.

**Figure supplement 2—source data 2.** Full raw unedited files of *Figure 7—figure supplement 2—source data 1*.

**Figure supplement 2—source data 3.** Quantification of vacuoles and cell death in hypotonic stress upon overexpression of point mutant ScDIP2 shown in *Figure 7—figure supplement 2F, G*.

---

subspecies are found to be toxic to yeast cells as their accumulation interfere with cellular processes like ER homeostasis and osmoadaptation. The unaltered levels of other lipid metabolites that induce ER stress for example, saturated FAs, sterols (*Volmer and Ron, 2015*), DAG precursors, that is, PA and lyso-PA (*Henry et al., 2014*), further validates that DAG–TAG imbalance is the cause of ER stress in the absence of DIP2. Overexpression of DIP2 in yeast increases the flux from DAG to TAG, resulting in massive depletion of DAG subspecies below the basal level. As we find that these DAG subspecies are required for vacuole membrane fusion, diminishing their cellular abundance upon DIP2 overexpression leads to decreased vacuole fusion and increased cell death upon osmotic stress in yeast. Thus, the study essentially identifies an unconventional and selective DAG metabolism route in yeast with important physiological consequences. Similar to the yeast results, the lipidomics analysis indicates that the DIP2-mediated DAG regulation in *Drosophila* and mice is also biased towards a subset of DAG species. Given the tissue-specific expression (*Nitta et al., 2017*; *Zhang et al., 2015*) and the presence of multiple paralogues of DIP2 in higher organisms, the specificity and the functional significance of DAG regulation by DIP2 need to be probed in further studies.

The study characterizes DIP2 as a non-canonical member of DAG metabolism pathways as it shares no homology with known DAG metabolizing enzymes of eukaryotes. Interestingly, DIP2 harbours a tandem FLD didomain which itself is a novel domain organization with respect to the evolutionary history of FLD. Although the canonical function of standalone FLD is to activate FAs, we find that the didomain architecture of FLD is essential for the DAG to TAG conversion by DIP2. We also provide evidence for the role of adenylation activity of FLDs of ScDIP2 in facilitating the conversion of DAG to TAG using mutations in the conserved adenylation motifs. This leads us to propose a putative biochemical scheme where DIP2 is involved in the fatty acylation of DAGs to produce TAGs, utilizing an activated acyl-adenylates intermediate. Despite significant efforts, we were unable to show the DAG to TAG conversion reaction *in vitro* using individual components along with purified protein. Although this marks the limitation of the current study, it highlights the involvement of other cellular factors or specific membrane environment necessary for the reaction. The involvement of known DAG acyltransferases (Dga1, Lro1, Are1, and Are2) in the DIP2-mediated DAG regulation pathway cannot be ruled out. However, the accumulation of the subset of DAGs in a DIP2 mutant despite the presence of those acyltransferases does suggest that DIP2-mediated pathway is independent of known TAG forming enzymes (*Figure 7H*).

We find that DIP2 uniquely localizes to mitochondria–vacuole membrane contact sites in yeast. Typically, a membrane contact site is a close apposition between membrane of two different organelles populated with structural proteins for tethering membranes and functional proteins for exchanging metabolites or ions (*Scorrano et al., 2019*). It is widely speculated that specific proteins are enriched at the contact sites to regulate the unique membrane compositions at contact sites (*Jain and Holthuis, 2017*). Although, recent studies have identified several tethering proteins at mitochondria–vacuole contact sites (*Elbaz-Alon et al., 2014*; *González Montoro et al., 2018*; *John Peter et al., 2017*), functional proteins at the sites remain largely unknown. Given the localization of DIP2 to a contact site and its specialized role in regulating DAG subspecies, the study uncovers DIP2 as a new family of functional protein at the mitochondria–vacuole contact site. Since DAG to TAG conversion via canonical pathways takes place at ER and lipid droplets (*Sorger and Daum, 2003*), a thorough investigation is needed to explore the role of unconventional DIP2-mediated DAG metabolism at mitochondria–vacuole contact sites.

Total DAG pool constitutes only a relatively small population (3–5%) within the lipidome (*Casanovas et al., 2015*; *Ejsing et al., 2009*) and is maintained via the rapid turnover to either storage lipids or membrane lipids. Indeed, the deletion of DAG metabolizing enzymes of yeast such as Dgk1, which facilitates the formation of phospholipids, or Dga1 and Lro1, that facilitate TAG synthesis results in the accumulation of bulk DAG species (*Fakas et al., 2011*; *Mora et al., 2012*; *Oelkers et al., 2002*; *Oelkers et al., 2000*; *Rockenfeller et al., 2018*). Surprisingly, unlike the DIP2 mutant which accumulates a subset of the DAG pool (<0.005%), the accumulation of bulk DAGs in Dga1 or Lro1 or Dgk1 mutants does not lead to ER stress induction. Similarly, specific TAGs related to these DAGs are only altered in the mutant leaving the bulk TAG pool almost unaffected. Since the levels of bulk TAG pools and sterol esters in ΔScDIP2 cells show minimal or no change, ScDIP2 is likely to have a limited effect on lipid droplet (LD) biogenesis process. These observations point towards functionally independent DAG populations, viz., a benign bulk DAG pool, utilized mainly for bilayer production and energy storage, and a minor context-dependent DAG pool. The present study identifies the minor DAG pool as being resistant to any kind of redirection that retains them in the membrane. These DAGs can only be titrated as a storage lipid, TAG, by DIP2 in the logarithmic growth phase and via oleic acid-mediated 'metabolic reset' in the stationary phase. Thus, the accumulation of these DAG subspecies is likely to impact the physical properties of the membrane, which subsequently results in UPR triggering and vacuole fusion dysregulation.

Overall, the current work shows that DIP2-mediated fine-tuning of a fraction of the DAG pool is required for adapting to different types of stress such as ER proteostasis and osmolarity. Unlike the laboratory conditions, these adaptation helps in the survival of yeast in the diverse environmental niches in nature (*Figure 7H*). As pathogenic fungi adapt to the changing microenvironments during disease progression, similar stress adaptation processes linked to DAG regulation becomes crucial for their survival inside the host. Studies on plant and animal pathogenic fungi have indeed suggested that DIP2 is involved in infection-related morphogenesis like conidia, appressoria, hyphae, and spherule formation (*Lu et al., 2003*; *Narra et al., 2016*; *Wang et al., 2016*), where vacuole fusion–fission dynamics plays a vital role (*Veses et al., 2008*). Consequently, DIP2 has been identified as a potential candidate for designing fungicides and developing a vaccine against valley fever (*Shubitz et al., 2018*). Furthermore, DAG regulation in neurons is found as a critical factor for neuron branching, dendritic spine formation (*Tu-Sekine et al., 2015*), etc., which are the major cellular processes affected in autism and related neurodevelopmental disorders (*Gilbert and Man, 2017*; *Phillips and Pozzo-Miller, 2015*). Since DIP2 has been recognized as a potential risk factor for ASD and other neurodevelopmental disorders (*Supplementary file 1*), aberrant accumulation of chemically distinct DAG subspecies is likely to be the potential molecular driver of these disorders. Interestingly, the previous reports suggesting the requirement FLD didomain of DIP2 for proper neuron branching in *Drosophila* and *C. elegans* (*Nitta et al., 2017*; *Noblett et al., 2019*) correlates with our observation on the need of FLD didomain in selective DAG to TAG conversion. This also suggests the possible involvement of DIP2 in driving these crucial developmental processes via DAG subspecies regulation. Therefore, the identification of DIP2 as a regulator of DAG subspecies abundance provides a solid platform to bridge the gap between its cellular function and phenotypes linked to disease pathogenesis. Further explorations will be focussed on (1) the elucidation of the biochemical and structural mechanism of DAG to TAG conversion by DIP2, (2) the relation between subcellular localization of DIP2 and its regulatory

role in defining the distribution of DAG subspecies at subcellular sites (*Baron and Malhotra, 2002*; *Cowell et al., 2009*; *Ganesan et al., 2019*; *Starr and Fratti, 2019*; *Yang and Kazanietz, 2003*), and (3) how misregulation of a fraction of DAG pool translates into a pathological manifestation.

## Materials and methods
### Yeast strains and plasmids

All strains used in this study are derivatives of *Saccharomyces cerevisiae* S288C and are listed in *Supplementary file 2*. Tagging of proteins and gene deletions were performed by standard PCR-based homologous recombination (*Longtine et al., 1998*). Strains used in this study are isogenic to BY4741 (Mata ura3Δ his3Δ1 leu2Δ0 met15Δ0) (*Brachmann et al., 1998*) . Briefly, ScDIP2 (Cmr2; SGD ID: S000005619) gene was replaced with a kanamycin resistance cassette, PCR amplified from pFA6A-KanMX6 plasmid to have flanking sequences homologous to the 5′ and 3′ end of ScDIP2 gene. The PCR product was purified and transformed through direct carrier DNA/PEG method-based transformations (*Gietz and Schiestl, 2007*) and ScDIP2 deletion mutant colonies (YSM57 and YSM70) were screened for kanamycin selection. Thus, obtained resistant colonies were screened and validated for deletion using PCR through the flanking sequences of kanamycin cassette as primers. YSM101 strain was obtained by C-terminally tagging ScDIP2 at genomic locus with a GFP-KanMX6 cassette amplified from pFA6a-GFP(S65T)-KanMX6 (*Longtine et al., 1998*). pYSM5 vector was generated by cloning GAL1 promoter and sequence of TEV-mEGFP-8XHis-2XHA in pLE124 vector, a kind gift from Dr. M Palani's laboratory. ScDIP2 full-length gene and domain-truncated sequences were cloned under GAL1 promoter and tagged C-terminally with TEV-mEGFP-8XHis-2XHA using restriction digestion and ligation cloning method. pYSM10 vector was generated by cloning ScDIP2 sequence into PacI–XhoI sites of pYSM5 vector. pYSM7 vector was generated by cloning ScDIP2 sequence with corresponding promoter and terminator sequence from the genomic locus in pPPM90 vector. Site-directed mutagenesis was performed by PCR using mutagenic oligonucleotides with pYSM10 as the template. Mutagenized plasmids were checked by sequencing. All the primers and plasmids used in this study are in *Supplementary file 3* and plasmids are listed in *Supplementary file 4*.

### Media and reagents

Yeast cells were grown at 30°C in a synthetic complete (SC) media. SC media contained 1.7 g/l yeast-nitrogen base with ammonium sulphate (BD Difco), 20 g/l glucose and was supplemented with the required amino acid mixtures (Sigma) or lacking respective amino acids or supplemented with G418 (200 µg/ml) for strains and plasmid selection. For some experiments, yeasts were grown in YP media (yeast extract – 20 g/l, peptone – 20 g/l) supplemented with 2% dextrose or galactose. Different yeast strains were stored at −80°C as glycerol stocks. Prior to all experiments, we briefly thawed the yeast stocks, and few microlitres of aliquot was streaked on a SC agar plate and grown under selection at 30°C for 2 days. Before the experiment, single colony was used to prepare seed cultures in 5–10 ml SC media or YP media grown under selection overnight at 30°C in a shaking incubator.

### *Drosophila* lines

The fly strains used in the study were procured from Bloomington *Drosophila* Stock Center, Indiana University, USA. DIP2 knock-out fly line from an earlier study (*Nitta et al., 2017*) was used and was a kind gift from Tetsuya Tabata, University of Tokyo. All stocks and their crosses were maintained in standard cornmeal–yeast–sugar–agar medium at 25°C under a 12–12 hr light–dark cycle in the Fly lab, CCMB.

### Cell lines

Generation of the MmDIP2A$^{-/-}$ line of mice with null alleles of the Dip2a gene was described previously (*Kinatukara et al., 2020*). The mouse embryonic stem cells (ESCs) used in the study were derived from blastocysts (*Behringer et al., 2014*) obtained from mating of MmDIP2A$^{-/-}$ mice (*Kinatukara et al., 2020*). The ESCs were authenticated by genotyping for *Dip2a* mutation and STR analysis by PCR (*Almeida et al., 2014*). Mycoplasma contamination was not detected in the cell lines using LookOut Mycoplasma PCR detection kit (Sigma). The obtained ESCs were grown as per previously described protocols (*Jana et al., 2019*). Briefly, cells were cultured on tissue culture-treated plates/

dishes coated with 0.1% gelatin, in GMEM media supplemented with L-glutamine, 100 µM β-mercaptoethanol, 1 mM non-essential amino acids (Gibco), 100 units/ml human LIF supplemented with 10% fetal bovine serum (Gibco). Cultures were grown in a humidified incubator at 37°C and 5% $CO_2$.

## Metabolic radiolabelling and lipid extraction

Yeast cells were inoculated at 0.05 $OD_{600}$ from an overnight grown primary culture in SC media. 0.5 µCi/ml 1-$^{14}$C acetic acid (American Radiolabeled Chemicals) was added to the media and grown up to mid-log phase ($OD_{600}$ ~3) to achieve steady-state labelling.

Lipids were extracted according to the Folch method (*Folch et al., 1957*). In brief, equal amounts of cells (10 ml log-phase culture, $OD_{600}$ – 3.0) were taken and washed with ice-cold 1× phosphate-buffered saline (PBS) and then resuspended in 250 µl of ice-cold PBS. Cell suspension was then transferred to a fresh glass vial and remaining cells in microfuge tube were rinsed with an additional 250 µl of PBS and added to the earlier collection. Acid-washed glass beads of 0.5 ml volume was added to the cell suspension, followed by the addition of 1.5 ml of chloroform ($CHCl_3$)/methanol (MeOH) mix (2:1 ratio). Cells were disrupted by shearing using high speed vortex for 10 min. Phase separation of solvents was achieved by centrifugation at 3000 rpm for 5 min. The organic phase at the bottom part was collected in fresh glass vials. Formic acid (Sigma) was added at 10% (vol/vol) to the aqueous phase and vortexed for 1 min. Again, 1 ml of $CHCl_3$ was added for re-extraction and vortexed for 5 min, followed by centrifugation and collection of bottom phase. The re-extraction step was repeated two times and the pooled organic phase was dried under nitrogen gas at room temperature (RT). Dried samples were stored at −80°C until further use.

## Thin-layer chromatography

Dried lipids were resuspended in $CHCl_3$:MeOH (1:1) and 10 $OD_{600}$ equivalent lipid was spotted on two separate silica gel TLC plates for neutral and polar lipid separation. Phospholipids were separated using solvent system-A ($CHCl_3$:ethanol:water:triethylamine; 30:34:8:35, vol/vol) (*Korte and Casey, 1982*). Neutral lipids were separated using solvent system-B (hexane:diethyl ether:glacial acetic acid; 80:20:2, vol/vol) (*Fakas et al., 2011*). Radiolabelled lipids were visualized by a phosphorimager (Amersham Typhoon FLA 9000). Densitometry analysis was performed using ImageJ2 software (*Rueden et al., 2017*) to measure relative intensities of labelled lipids. The percentages presented for the individual lipid species were normalized to the total $^{14}$C-labeled fraction.

## Lipidomics sample preparation

The lipid extraction process for different biological samples (yeast, *Drosophila*, and mouse ES cells) were performed using modified-Folch method as reported previously (*Abhyankar et al., 2018*; *Kelkar et al., 2019*; *Kumar et al., 2019*; *Pathak et al., 2018*). Briefly, yeast cell pellet harvested from 20 ml culture was washed and suspended in chilled PBS. Cells were disrupted by sonication at 60% amplitude for 1 s ON/ 3 s OFF for 10 cycles with a probe sonicator on ice bed. At this stage, 10 µl of lysates from each sample were collected for protein estimation that is required for data normalization during analysis. Then, $CHCl_3$ and MeOH mix with respective internal standards was added to lysate to achieve the ratio of 2:1:1 ($CHCl_3$:MeOH:PBS). Mixture was vortexed thoroughly and centrifuged at 3000 rpm at RT to achieve phase separation. Organic bottom phase was collected in fresh glass vials and 100 µl of formic acid (~10%, vol/vol of lysate volume) was added to remaining top phase. After brief vortexing, same volume of $CHCl_3$, as mentioned in previous step, was added for re-extraction. Previous steps were repeated to phase separate the mixture, bottom phase was pooled and dried under stream of $N_2$ at RT. Protein estimation was performed using Bradford assay reagent (Sigma).

For *Drosophila* samples, 5-day-old adult *Drosophila* were collected, flash frozen in liquid $N_2$ and stored at −80°C until further processing. For lipid extraction, six flies (three males and three females) were taken and crushed thoroughly in cold PBS with a hand-held homogenizer. Further steps were performed as per the protocol mentioned above.

Mouse ES cells were derived from wild type and MmDIP2A$^{-/-}$ lines as described previously were maintained in GMEM media (supplemented with LIF). ~8 million cells were washed with PBS and flash frozen in liquid $N_2$ before storing at −80°C until further processing. For lipid extraction, cells were resuspended in ice-cold PBS and disrupted using bath sonicator (1 s ON/ 3 s OFF for five cycles). Later the lipidome was extracted as per the protocol mentioned in the above.

## Lipidome analysis by mass spectrometry

The extracted lipids were re-solubilized in 200 µl of 2:1 (vol/vol) CHCl$_3$/MeOH, and 20 µl was used for the lipidomics analysis. The extracted lipids were analysed on a Silica F254 thin-layer chromatography and diethyl acetate:hexane:glacial acetic acid (20:80:1) was used as the mobile phase (*Hutchins et al., 2008*). The lipidomic analysis of the isolated lipid species was carried according to previously described protocols (*Abhyankar et al., 2018*; *Kelkar et al., 2019*; *Kumar et al., 2019*; *Pathak et al., 2018*). Lipid species were analysed and quantified using the multiple reaction monitoring high-resolution (MRM-HR) scanning method on a Sciex X500R QTOF mass spectrometer (MS) fitted with an Exion-LC series UHPLC with minor modifications. All data were acquired and analysed using the SciexOS software. The lipid species were quantified by measuring the area under the curve and the normalized raw MS intensities were used for comparison between various experimental groups. All the lipidomics data are represented as mean ± standard error of the mean (SEM) of six replicates.

## Intracellular glycerol measurement

Intracellular glycerol concentration was measured as described previously (*Dunayevich et al., 2018*). Yeast cells were grown in SC media, inoculated from an overnight grown culture, till the mid-log phase. An aliquot of 1 ml culture was taken and OD$_{600}$ was measured for all the replicates. The cells were harvested by centrifuging at 3000 rpm for 5 min at RT and washed twice with PBS. Cells were resuspended in 1 ml of boiling water and incubated at 100°C for 10 min. Next, the resuspension was cooled on ice for 10 min and centrifuged at 15,000 × *g* for 5 min. The supernatants were used to determine glycerol concentration using a commercial kit following the manufacture's indications (MAK117, Sigma). The standard curve was measured between 0 and 1 mM of glycerol. Samples were diluted as necessary to obtain data in this range. The amount of glycerol was normalized to OD$_{600}$ and the mean value of intracellular glycerol concentration/OD$_{600}$ ± standard deviation SD ($n=4$) was plotted.

## Serial dilution assay

Yeast cells were grown to a final OD$_{600}$ of 0.2 and used as the first dilution. The first dilution stock was used to generate multiple serial dilution stocks ($10^{-1}$, $10^{-2}$, $10^{-3}$, and $10^{-4}$). 5 µl cell suspension from each dilution stock were spotted sequentially on respective SD agar plates and grown for 48–72 hr at 30°C incubator, before imaging. For different stress assays, chemical reagents were added to nutrient agar media as per the indicated concentrations. For chemical supplementation assay, FAs dissolved in DMSO at different molar concentrations were used. Other phospholipid precursors like choline and ethanolamine were dissolved in autoclaved water and used for supplementation assays.

## UPR reporter assay

pPM47 (UPR-RFP CEN/ARS URA3) was a gift from Feroz Papa (Addgene plasmid # 20132). The experiment was performed according to the procedures described earlier (*Merksamer et al., 2008*). Briefly, cells harbouring single-copy reporter plasmid was grown in respective media up to early-log, mid-log phase, or stationary phase, harvested by centrifugation and washed with PBS before experiment. Fluorescence signal was measured with a flow cytometer, BD LSR Fortessa (excitation – 561 nm, emission – 610 nm, volume – 100 µl, number of cells – 10,000). Cells without reporter plasmid served as a control. To validate the UPR reporter assay, UPR signal was measured using cultures with increasing concentration of tunicamycin added and incubated for 2 hr at 30°C with shaking. For chemical supplementation, oleic acid was added to media at 1 mM concentration and grown up to log phase before subjecting for fluorescence measurement. Data analysis was performed using FCS Express (De novo software).

## Growth curve and cell viability assay

The growth curve was obtained by growing cells from an equal initial OD$_{600}$ of 0.2. Cultures were allowed to grow in 30°C incubator shaker and the OD$_{600}$ was recorded every 4 hr. These experiments were repeated in triplicates and the error bars represent SD values.

For hypertonic stress cell viability assay, overnight grown cells were diluted to OD$_{600}$ of 0.2 in SC-dextrose media and grown for 4–5 hr in 30°C incubator shaker. Log-phase cells were harvested, diluted to OD$_{600}$ of 0.2 and grown in presence of 1 M NaCl in YP-dextrose media. After 4–5 hr of

growth, cells were incubated with 0.2% (vol/vol) trypan blue (Sigma) for 5 min and further washed five times with PBS. Cell viability was assessed using bright field microscopy by counting the proportion of live cells (unstained) to dead cells (stained). Cells grown without NaCl was used as control.

For hypotonic stress cell viability assay, overnight grown primary cultures were diluted in SC-galactose media at $OD_{600}$ of 0.2. After growing till log phase, cells were diluted in 5 ml YP-galactose media with 1 M NaCl and incubated for 15 min at 30°C with 200 rpm shaking. Cells were harvested by centrifugation and re-dispersed in 5 ml autoclaved filtered Milli-Q (hypotonic condition), After incubation for 15 min at 30°C with 200 rpm shaking, cell death was assessed as per the protocol mentioned above. Cells incubated without salt and hypotonic condition were used as control.

## Live cell microscopy

ScDIP2-GFP knock-in strain was grown overnight in 5 ml of SC broth at 30°C. 20 nM of MitoTracker Red CMXRos (Invitrogen Inc) was added to these overnight grown cells and incubated further for 30 min with shaking. The MitoTracker incubated cells were then washed in autoclaved water. 5 µl of those cells were immobilized on a poly-L-lysine coated glass slides and visualized under a Zeiss Axio Imager.Z2, inverted widefield fluorescence microscope equipped with a HAL 100 illuminator, Plan Apochromat ×100 oil objective (NA 1.4), and a AxioCam CCD camera. Images were captured in DIC (Nomarski optics), Texas red (for MitoTracker Red) and eGFP fluorescence mode. The vacuole of ScDIP2-GFP knock-in cells was stained with FM4-64 dye (Invitrogen) following the protocol described in 'Vacuole morphology assay'. Mitochondrial DNA was stained with DAPI by treating the live cells (at mid-log growth phase) at a final concentration of 1 µg/ml DAPI directly in SC media (*Dellinger and Gèze, 2001*; *Higuchi-Sanabria et al., 2016*; *Zarin et al., 2021*). Cells were stained for 10 min shaking at 30°C, washed five times with SC media, and imaged immediately for no longer than 10 min.

Cells were further processed for slide preparation, and images were collected in same procedure described above. All the experiments were done in biological triplicates. For colocalization study, stacks of five images with a step size of 0.5 µm were collected and deconvolved using iterative mode in Zen 2.6 software. The colocalization between GFP and other fluorescence signals/puncta was calculated manually and with JACoP plugin (*Bolte and Cordelières, 2006*) in ImageJ2 software.

Sec63-mRFP (pSM1960, a gift from Susan Michaelis; Addgene plasmid # 41842) (*Metzger et al., 2008*) was used as ER marker. For nuclear staining, cells were fixed with 1% formaldehyde followed by incubation with 1 µg/ml DAPI (Sigma) in PBS for 10 min without shaking and subsequent washing with PBS. All images were processed using ImageJ2 software.

## Vacuole morphology assay

Vacuole morphology was determined by FM4-64 staining of the vacuole membrane as described previously (*Vida and Emr, 1995*). Briefly, overnight grown culture was inoculated in a fresh medium with and grown up to early-log phase ($OD_{600}$ – 1.2–1.5). Cells equivalent to 1.5 $OD_{600}$ were harvested by spinning at 4000 rpm for 5 min, washed with PBS and resuspended in 200 µl of media. Cells were pulse labelled with 10 µM of FM4-64 for 30 min at 30°C, then washed in medium without dye and finally suspended in 5 ml media to grow for 90 min at 30°C, 200 rpm. Cells were then harvested by centrifugation, washed twice with PBS and resuspended in PBS or SC media for microscopy. Diameter of the largest vacuole of budding yeast cell was measured manually using the line tool of ImageJ. Vacuolar diameters for indicated strains were plotted into box and whisker plots.

For vacuole fission–fusion assay (*Jones et al., 2010*), yeast vacuoles were stained first according to the abovementioned protocol. It is now well established that hypotonic stress induces vacuole fusion while hypertonic stress induces vacuole fission *in vivo*. Therefore, cells were subjected to the hypertonic condition, that is, YP-dextrose or YP-galactose with 0.4 M NaCl and grown for 10–15 min at 30°C, 200 rpm. The vacuole fission was confirmed by visualizing under fluorescence microscope. To induce vacuole fusion, cells were harvested and subjected to hypotonic condition, that is, 10-fold dilution in autoclaved Milli-Q water and grown for 10 min, 30°C, shaking at 200 rpm. Images were captured by putting washed cells on 2% low-melting agar bed on glass slides and mounted with a cover slip.

## Western blot analysis

For GFP-tagged full-length and domain-truncated constructs of ScDIP2 expressing strains, the cell pellet was resuspended in 20% trichloroacetic acid, mixed with ice cold glass beads (0.5 mm diameter) and ruptured by bead beating for 3 cycles (1 min ON, 1-min OFF). The supernatant was collected, and the pellet was harvested by centrifugation at 14,000 rpm for 10 min. Pellet was washed with ice-cold acetone, then dried under vacuum and finally dissolved in SDS-loading buffer. All the steps were performed at 4°C. Protein extracts were separated by Sodium dodecyl sulphate–polyacrylamide gel electrophoresis (SDS–PAGE). After separation, proteins were transferred onto a nitrocellulose membrane by wet transfer method at 100 V for 2 hr. Membranes were blocked with 5% non-fat milk powder in TBST buffer (20 mM Tris–HCl pH 7.5, 150 mM NaCl, 0.1% Tween-20) and immunoblotting was performed using anti-Pgk1 (monoclonal, Thermo Fisher Scientific) and anti-GFP (monoclonal, Cell Signaling Technologies). Signals were developed using the BioRad ChemiDoc MP imaging system.

## Isolation of total membrane

Total endomembrane fraction was isolated as described previously (*Sorin et al., 1997*). ScDIP2-GFP knock-in cells were resuspended in a cell resuspension buffer (50 mM Tris/HCl, pH 7.6, 0.6 M sorbitol, 1 mM Ethylenediaminetetraacetic acid (EDTA)) containing protease inhibitor cocktail (Calbiochem) and 1 mM PMSF (Sigma). Cells were disrupted by vortexing the cell suspension with acid-washed chilled glass beads for 8 cycles of 30 s pulse and 30 s chilling on ice. Lysate was centrifuged at 1000 × *g* for 10 min to separate cell debris as a pellet and collect the supernatant that contains organelles, membranes, and cytoplasmic proteins. The crude membrane fraction was collected by centrifuging the supernatant at 100,000 × *g* for 1 hr at 4°C and further resuspending the pellet in membrane resuspension buffer (20 mM Tris/HCl, 0.3 M sucrose, and 0.1 mM CaCl$_2$). The fractions were subjected to SDS–PAGE and ScDIP2-GFP was probed with anti-GFP antibody as per protocol mentioned in the above section.

## Bioinformatics analysis

All sequences were identified and retrieved from the NCBI sequence database using ScDIP2 (UNIPROT ID: Q12275) as a template and BLAST search algorithm (*Altschul et al., 1990*). Domain boundaries of ScDIP2 were marked based on consensus obtained from structure-based sequence alignments generated in EXPRESSO (*Armougom et al., 2006*), manual inspection of homology models, generated via MODELLER (*Webb and Sali, 2016*), visualized in PyMOL (*Schrödinger, 2022*), and domain definitions obtained from Conserved Domain Database (*Lu et al., 2020*). Multiple sequence alignment was generated using MAFFT (*Katoh et al., 2002*) and MUSCLE (*Edgar, 2004*). The sequence alignments were rendered using ESPript 3.0 (*Robert and Gouet, 2014*). The phylogenetic analysis of these sequences was carried out using maximum-likelihood method as implemented in IQTREE (*Nguyen et al., 2015*) using default parameters. The phylogenetic tree was rendered in iTOL server (*Letunic and Bork, 2021*). The images of various organisms were reusable silhouette images of organisms obtained from PhyloPic (http://www.phylopic.org/), under Creative Commons license. Pairwise sequence identity was calculated using SIAS (http://imed.med.ucm.es/Tools/sias.html) with default parameters. Sequence logos were generated using WebLogo (*Crooks et al., 2004*). Homology models for structural analysis were obtained from AlphaFold Protein Structure Database (*Jumper et al., 2021*; *Varadi et al., 2022*).

The GO terms were tabulated by analysing the genetic interactions using DRYGIN database (https://thecellmap.org/) (*Koh et al., 2010*) at default stringency levels (between +0.08 and −0.08) along with tools from YeastMine (*Balakrishnan et al., 2012*) and Gene Ontology Slim Term Mapper (*Ashburner et al., 2000*).

## Statistics

Error bars represent SEM or SD as indicated in the respective figure legends. Data were processed and analysed using Microsoft Excel and GraphPad Prism v.8.0.2.263. Statistical analysis of the differences between two groups was performed using a two-tailed, unpaired *t*-test. Based on the p value obtained, the significance of differences marked as *p < 0.05; **p < 0.01; ***p < 0.001; ***p < 0.0001; ns, not significant.

## Acknowledgements

We thank Palani Murugan Rangasamy (CSIR-CCMB) for helping in generation of knock-out strains of yeast and for sharing plasmids; Animal house facility, Fly lab facility, Advanced Microscopy facility and Flow cytometry (FACS) facility for help in performing experiments. We thank Krishnaveni Mishra (UoH) for sharing yeast knock-out strains and Rakesh Mishra (CSIR-CCMB) for helping with *Drosophila* experiments. We thank Koustav Sanyal (JNCASR) and Hashim Reza (JNCASR) for insightful discussion on vacuole morphology; Rajesh Gokhale (NII), Durgadas P Kasbekar (CDFD) and Venkat R Chalamcharla (CSIR-CCMB) for their valuable comments on the manuscript. SM thank CSIR, India and PK thank the Department of Biotechnology, India for research fellowship. SaS thanks UGC, India for research fellowship. SSK thank DBT/Wellcome Trust India Alliance Fellowship (grant number IA/I/15/2/502058) and a Department of Science and Technology (DST) Fund for Improvement of S&T Infrastructure (grant number SR/FST/LSII-043/2016) to the IISER Pune Biology Department. RS thank NCP under health care theme project of CSIR, India; J.C. Bose Fellowship of SERB, India; and Centre of Excellence Project of Department of Biotechnology, India.

## Additional information

### Competing interests

Rajan Sankaranarayanan: Reviewing editor, *eLife*. The other authors declare that no competing interests exist.

### Funding

| Funder | Grant reference number | Author |
| --- | --- | --- |
| Council for Scientific and Industrial Research (CSIR), India | | Sudipta Mondal |
| Department of Biotechnology, Ministry of Science and Technology, India | | Priyadarshan Kinatukara |
| University Grants Commission | | Sakshi Shambhavi |
| DBT/Wellcome Trust India Alliance Fellowship | IA/I/15/2/502058 | Siddhesh S Kamat |
| Department of Science and Technology, Ministry of Science and Technology, India | SR/FST/LSII-043/2016 | Siddhesh S Kamat |
| J.C. Bose Fellowship | | Rajan Sankaranarayanan |
| NCP under health care theme project | | Rajan Sankaranarayanan |
| Centre of Excellence Project | | Rajan Sankaranarayanan |

The funders had no role in study design, data collection, and interpretation, or the decision to submit the work for publication. For the purpose of Open Access, the authors have applied a CC BY public copyright license to any Author Accepted Manuscript version arising from this submission.

### Author contributions

Sudipta Mondal, Priyadarshan Kinatukara, Conceptualization, Data curation, Formal analysis, Investigation, Methodology, Validation, Visualization, Writing - original draft, Writing - review and editing; Shubham Singh, Data curation, Formal analysis, Investigation, Methodology, Validation, Visualization, Writing - review and editing; Sakshi Shambhavi, Formal analysis, Investigation, Methodology,

Validation, Visualization, Writing - review and editing; Gajanan S Patil, Formal analysis, Methodology, Validation, Visualization, Writing - review and editing; Noopur Dubey, Formal analysis, Investigation, Visualization, Writing - review and editing; Salam Herojeet Singh, Formal analysis, Methodology, Writing - review and editing; Biswajit Pal, Data curation, Formal analysis, Validation, Writing - review and editing; P Chandra Shekar, Data curation, Methodology, Validation, Writing - review and editing; Siddhesh S Kamat, Data curation, Formal analysis, Methodology, Supervision, Validation, Writing - review and editing; Rajan Sankaranarayanan, Conceptualization, Data curation, Formal analysis, Funding acquisition, Methodology, Project administration, Supervision, Validation, Visualization, Writing - original draft, Writing - review and editing

### Author ORCIDs
Sudipta Mondal http://orcid.org/0000-0002-3923-7449
Priyadarshan Kinatukara http://orcid.org/0000-0003-2210-2369
Sakshi Shambhavi http://orcid.org/0000-0002-8852-1542
Siddhesh S Kamat http://orcid.org/0000-0001-6132-7574
Rajan Sankaranarayanan http://orcid.org/0000-0003-4524-9953

### Ethics
This study was carried out in strict accordance with the recommendations in the Committee for the Purpose of Control and Supervision of Experiments on Animals (CPCSEA), India. The protocols were approved by the Institutional Animal Ethics Committee (IAEC) of CSIR-Centre for Cellular and Molecular Biology, Hyderabad, India (20/ GO/RBi/99/CPCSEA). All terminal experiments were performed by cervical dislocation after anesthetizing with isoflurane. All efforts were made to minimize suffering during all experiments.

### Decision letter and Author response
Decision letter https://doi.org/10.7554/eLife.77665.sa1
Author response https://doi.org/10.7554/eLife.77665.sa2

## Additional files

### Supplementary files
• Supplementary file 1. Tabulation of DIP2 associated phenotypes and disorders in different organisms.
• Supplementary file 2. List of yeast strains.
• Supplementary file 3. List of primers.
• Supplementary file 4. List of plasmids.
• Transparent reporting form

### Data availability
All data generated or analysed during this study are included in the manuscript and supporting file; Source Data files have been provided for all figures in the manuscript.

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
