## [Editor Report]

This manuscript reports that a previously uncharacterized protein, DIP2, which is localized at mitochondria and mitochondria-vacuole contacts is involved in the metabolism of a subset of diacylglycerol species. This study would be of interest to biologists interested in lipids, membrane contacts, metabolism, and ER stress.

---

## [Decision Letter]

**Decision letter after peer review:**

Thank you for submitting your article "DIP2 is a unique regulator of diacylglycerol lipid homeostasis in eukaryotes" for consideration by *eLife*. Your article has been reviewed by 3 peer reviewers, including Felix Campelo as the Reviewing Editor and Reviewer #1, and the evaluation has been overseen by Vivek Malhotra as the Senior Editor.

The reviewers have discussed their reviews, and the Reviewing Editor has drafted the following summary to help you prepare a revised submission.

Essential revisions:

1) Further characterize the actual role of DIP2 in lipid metabolism:

– It is not clear from the data whether the changes observed (more DAG, less TAG in the KOs; and opposite in the o/expression) are due to changes in lipolysis or in synthesis. The authors should measure changes in DAG precursors (e.g. PA, lyso-PA) (see Rev. #3, point #2).

– Although the authors already state in their paper that *in vitro* tests were not successful, it would be important that the authors can provide some more insight into whether the role of DIP2 in DAG to TAG conversion is direct or indirect (see Rev. #1, point #3; Rev. #2)

2) The authors should provide stronger evidence for the role of DIP2 in vacuoles.

– Provide better images for the localization of DIP2 (see Rev. #1, point #1; Rev. #2; Rev. #3, point #4).

– Further test the role of DIP2 in vacuoles (is it direct or indirect?) (see Rev. #1, point #2and2b).

As an optional possibility, it would also be interesting to test the impact of DIP2 loss in LD biogenesis and shape. Moreover, going further into the mechanism by which DIP2 has a specific role only in a subset of DAGs can be very interesting, but I feel that perhaps falls a bit beyond the scope of this paper. The authors can of course discuss it a bit further.

*Reviewer #1 (Recommendations for the authors):*

I see two weak points in this manuscript:

1) The fluorescence microscopy data showing localization of the DIP2 protein to vacuole/mitochondria could be very much improved.

2) The vacuole fusion defect in the absence of DIP2 is quite interesting and promising. However, I feel like further data would be required to fully support such a claim. For instance, could the authors acutely remove DIP2 (anchor-away, degron system, etc.) to then perform time-lapse imaging to show, as predicted by the authors, vacuole fusion?

(2b) Related to 1 and 2, would the authors have a way to monitor the lipid changes in specific organelles, such as vacuole or mitochondria?

3) Is DIP2 indeed a lipid metabolizing enzyme or is it a regulator/modifier of other metabolizing enzymes? I appreciate that the authors claim and write in their manuscript that they have not been able to show *in vitro* the capacity of DIP2 to convert some DAG species into TAG. How would the authors interpret such negative results? Because of experimental/technical limitations or because indeed DIP2 is not able to convert DAG *in vitro*?

*Reviewer #2 (Recommendations for the authors):*

The authors should provide *in vitro* evidence of DIP2 activity (using recombinant proteins encompassing or not the FLDs and their mutants) and address the issue of its specificity at the molecular detail. The localization of ScDIP2 is reported to be vacuolar + mitochondrial. The images shown by the authors would point towards a primarily mitochondrial localization. This aspect needs to be investigated further (and possibly expanded to mammalian and DmDIP2) in light of the proposed DIP2 biology.

*Reviewer #3 (Recommendations for the authors):*

1) As mentioned above, some DAG lipids go up in ScDIP2-ko. What is the effect on precursor lipids that make DAG, such as PA and lyso PA? Changes in these lipids can also influence ER homeostasis and ER stress.

2) the decrease in TAG in the ScDIP2 knockout could be due to an increase in lipolysis or a decrease in TAG synthesis. Here, increased lipolysis is ruled out because there is no significant increase in MAG or FFAs in this KO (Figure 2, figure supplement 2B,c). This conclusion would be more strongly supported if the rate of glycerol release could be monitored in control and ScDIP2 KO. MAG and FFAs can often be rapidly re-incorporated into other lipid species, or simply oxidized. Thus, the lack of their accumulation alone does not seem sufficient to rule out elevated lipolysis in the KO.

3) Loss of ScDIP2 induces ER UPR stress, which can be rescued by treatment with oleic acid (Figure 3C). Oleic acid treatment drives lipid droplet biogenesis in yeast, which is commonly associated with releasing various ER stresses. Does this rescue require LD biogenesis? Relatedly, how does loss of ScDIP2 impact LD biogenesis and morphology?

4) Through primarily co-imaging with ScDIP2-GFP and vacuole strain FM4-64, it is proposed that a pool of SCDIP2-GFP localizes to the vacuole surface. As presented in Figure 5A, this ScDIP2-GFP does not look vacuolar, but possibly localized to mitochondria adjacent to the vacuole. Some additional evidence supporting vacuole localization would strengthen this observation. Potentially blotting for ScDIP2 on isolated vacuoles.

[Editors' note: further revisions were suggested prior to acceptance, as described below.]

Thank you for resubmitting your work entitled "DIP2 is a unique regulator of diacylglycerol lipid homeostasis in eukaryotes" for further consideration by *eLife*. Your revised article has been evaluated by Vivek Malhotra (Senior Editor) and a Reviewing Editor.

The manuscript has been improved but there are some remaining issues that need to be addressed, as outlined below:

We all agree that this study is generally well done and that it constitutes a potentially exciting piece of work. However, upon consultation amongst reviewers, we still think that the interpretation of the new microscopy data as a proof for vacuolar localization of ScDIP2 is not conclusive. An alternative explanation of the data could be that all ScDIP2 localizes to mitochondria, and that the vacuole association (as observed by fluorescence microscopy) represents mitochondrial-vacuole contacts.

In order to clarify better this point, we request that you perform 3-color imaging of DIP2 together with mitochondrial and vacuolar markers, which will help reveal whether the association between DIP2 and vacuoles as observed in the present version of the manuscript corresponds to mitochondrial-vacuolar contacts or not. These data could even provide an exciting new observation if it confirms mitochondria-vacuole contacts, since there are not that many proteins known to localize to these contacts. The authors should adjust the text according to the outcome of these experiments.

*Reviewer #1 (Recommendations for the authors):*

Although the authors, for different reasons, have not been able to perform some of the requested experiments, they have clarified the paper and tempered the claims, so now I think this is a solid paper that deserves being published at eLife.

*Reviewer #2 (Recommendations for the authors):*

The authors sufficiently addressed my concerns. I, therefore, support publication in eLife

*Reviewer #3 (Recommendations for the authors):*

The study investigates the role of DIP2 enzymes primarily in yeast, and their roles in neutral lipid metabolism. In general the study is well conducted and provides significant preliminary data on the role of these enzymes in DAG:TAG lipid metabolism. The only major remaining concern is the conclusion of the vacuole localization of ScDIP2. Genetic manipulation of ScDIP2 clearly affects vacuole morphology and fusion, etc, but the fluorescent imaging data, as presented, does not conclusively show vacuole localization of ScDIP2-GFP. An alternative model is that all ScDIP2 simply localizes to mitochondria, and that the vacuole association simply represents mito-vacuole contacts. The new imaging data presented in this revised manuscript are compelling, but cannot rule out this alternative model. Since this vacuole localization appears to be a major tenet of the study's conclusions, additional evidence such as biochemical analysis, or even imaging with dual mito and vacuole markers, is necessary prior to making such conclusions.

---

## [Author Response]

Essential revisions:1) Further characterize the actual role of DIP2 in lipid metabolism:– It is not clear from the data whether the changes observed (more DAG, less TAG in the KOs; and opposite in the o/expression) are due to changes in lipolysis or in synthesis.

We thank the reviewer for presenting an alternate hypothesis for explaining why diacylglycerols (DAGs) have accumulated while triacylglycerols (TAGs) have reduced in the DIP2KO mutant. It allows us to further clarify based on our results that show that DIP2 knockout does not drive DAG-TAG imbalance through lipolysis.

As per the alternative hypothesis, enhanced lipolysis of TAGs could have resulted in the formation of DAGs and depletion of TAGs with a concomitant release of free fatty acids (FFA). If further lipolysis takes place, monoacylglycerols (MAGs) will be formed that will eventually be converted to FFA along with glycerol (see Author response image 1).

**Author response image 1. sa2fig1:** A schematic showing lipolysis of TAGs to form DAGs, MAGs, FFA and Glycerol.

Since ScDIP2 KO accumulates only DAGs without any change in MAG and FFA levels (Figure 2—figure supplement 2B and 2C), the lipolysis will have to be limited to conversion of TAGs to DAGs and not extend forward. In such a scenario, we will not see any change in glycerol levels. However, if for some reason the lipolysis further continues, there will be three-fold molar excess of FFA than glycerol as shown in the schematic. We do agree that reutilization of MAGs and FFA is greater than glycerol. However, it has been noted earlier that a 1:1 ratio of FFA to glycerol is maintained under homeostatic conditions, which increases to 3:1 under lipolytic-stimulated conditions (Schweiger et al., 2014).

Therefore, while measuring glycerol levels to see if a lipolytic defect may be possible, measuring FFA will be an equally sensitive readout. Also, early studies for characterization of lipases from yeast have also relied on measuring MAGs and FFAs (Heier et al., 2010; Kurat et al., 2006; Selvaraju et al., 2016) without invoking glycerol measurement. Therefore, based on the above discussion, we suggest that a change in MAG and FFA levels should suffice to draw a conclusion.

To further strengthen our conclusions, we were planning to carry out quantification of glycerol levels, as suggested by the reviewer. Unfortunately, the widely used glycerol assay kit ( MAK117-1KT) has stock unavailability due to COVID crisis and presently we are unable to perform the assay. We will perform the same in future, as and when the kit will be available. We now tone down our results by specifically mentioning that ‘In order to further completely rule out the role of lipolysis, glycerol level also can be measured’ (line# 152-158) in the revised manuscript.

– The authors should measure changes in DAG precursors (e.g. PA, lyso-PA) (see Rev. #3, point #2).

We agree with the reviewer that imbalance in the immediate precursors of DAGs such as phosphatidic acid (PA) and lysophosphatidic acid (LysoPA) (see Author response image 2) is also known result in phenotypes such as ER stress.

**Author response image 2. sa2fig2:** A schematic showing the lyso-PA and PA as the precursor of DAG synthesis.

As per the suggestion, we have now also included lysoPA levels along with PA levels (see Figure 2—figure supplement 1), which also do not show differences in levels across chain lengths. The PA levels were measured and already presented in the earlier version of the manuscript as Figure 2—figure supplement 1G. Therefore, it further strengthens our argument that the observed ER-stress is not a result of imbalance in precursors of DAGs and solely coming from the accumulation of specific DAG species.

This data is now added to the manuscript as Figure 2—figure supplement 1I. A summary of these experimental observation in also included in the main text at line# 367-370 of the revised manuscript.

– Although the authors already state in their paper that *in vitro* tests were not successful, it would be important that the authors can provide some more insight into whether the role of DIP2 in DAG to TAG conversion is direct or indirect (see Rev. #1, point #3; Rev. #2)

We thank the reviewer to help us explain an important aspect of the paper which concerns the role of DIP2 as a direct or indirect DAG to TAG converter. We agree that in the absence of any direct evidence, as pointed out by the reviewers, DIP2 may not be an actual enzyme and instead be simply a regulator or acts in coordination with other metabolizing enzymes. However, we summarize here all the evidence that clearly point towards a catalytic role for DIP2 in DAG to TAG conversion.

1. We find that lipidome is unaltered except for specific DAGs and TAGs in DIP2 deletion mutant. The domain truncation mutations of DIP2 show that the didomain of FAAL-like domains (FLD) is the minimal functional unit for facilitating DAG to TAG conversion. Further, we also demonstrate that within this minimal functional unit, a couple of important residues, which are known to be essential for catalysis in the superfamily to which the FLD belongs, are the key players. A schematic is shown in **Author response image 3**Author response image 3 to summarize the above-mentioned points.

**Author response image 3. sa2fig3:** 

2. Numerous crystal structures of such mutant proteins of various members this superfamily shows that mutations at these sites does not alter protein structure. This has been shown in multiple systems that theses mutations render the enzyme inactive by blocking a specific process of the catalytic mechanism and not because of structural collapse. Moreover, the site resides deep within the protein structure and is not surface exposed at any stage during the different steps of catalysis. We used a protein model generated by AlphaFold (Jumper et al., 2021) to illustrate how buried the residue is within the structure (see Author response image 4).

**Author response image 4. sa2fig4:** A sausage representation of the homology model of the FAAL-like domain-1 shows the overall residue depth (Chakravarty and Varadarajan, 1999; Tan et al., 2013). The core regions appear thicker than the surface regions, which are also coloured accordingly. Residues deep within the core are coloured red, while those at the surface are coloured blue. This representation shows that the mutated residues, D523 and L687 (spheres), are part of the core and deeply buried. A scatter plot showing the depth (Y-axis) of each residue (X-axis) in the region of interest, also shows that the residues are deeply buried in the core (3^rd^ quartile or > 75% percentile).

Therefore, the possibility that the mutations resulted in a structural collapse that further disturbed the interaction of DIP2 with other proteins can be ruled out. Thus, it can be concluded that ensuring homeostasis in the levels of specific DAG and TAG species is directly connected to a few conserved residues in its active site suggesting the direct role of DIP2 catalysis in DAG to TAG conversion.

3. Based on the available knowledge of lipid biosynthetic pathways, analyses of lipid species upstream and downstream to these nodes (DAG and TAG) argues for a defective or downregulated acyltransferase reaction in a DIP2 deletion mutant as well as the above-mentioned point mutant of DIP2.

4. Incidentally, the altered lipid species are highly under-represented in the lipidome (< 0.005 %), which is in line with earlier observations made from large-scale lipidomic studies in yeast (Casanovas et al., 2015; Ejsing et al., 2009). Complementation experiments show that DIP2 or the didomain of FAAL-like domains have remarkable chain-length restriction and limit their effect to some of the very rare DAG species in the lipidome. Even upon overexpression using a galactose-inducible promoter, only C36:0, C36:1 DAGs and related TAG show changes while the other species show minimal to no changes. The non-physiological levels of DIP2 resulting from galactose-induced overexpression shows minimal nonspecific targeting of DAGs and TAGs, while in the case of native promoter-induced expression only the cognate DAGs and TAGs are affected. It is further supported by the fact that under native conditions, DIP2 maintains the DAG and TAGs to basal level, while galactose-induced over-expression decreases DAGs and TAGs below the basal level. It should also be noted that such specificity is rarely encountered in lipid biosynthetic enzymes of any organism, particularly any TAG-forming acyltransferase.

These observations are indicative of a specific role for DIP2 in the DAG to TAG reaction. Having said this, we are yet to demonstrate the DAG to TAG conversion *in vitro*, as mentioned in the manuscript. We have tried to assess the formation of adenylation using radio-TLC as previously described (Patil et al., 2021) and to measure the formation of triacylglycerol using the DGAT assay (Oelkers et al., 2002). The positive controls for these experiments, viz., FACLs producing acyl-adenylates and Dga11 catalysing the formation of TAGs from DAGs, suggest that conditions are suitable for catalysis. However, the purified DIP2 could not generate any of these products. Based on these observations, possible biochemical function of DIP2 can only be speculative. As an alternative, DIP2 could be a sensor that directly or indirectly facilitates the conversion of specific DAGs to TAGs. It may also be possible that DIP2 is an accessory enzyme necessary for an acyltransferase that selectively converts specific DAGs to TAGs. Both these alternative hypotheses require DIP2 to rely on an existing acyltransferase that can drive the conversion of specific DAGs to TAGs. There are only four known acyltransferases in yeast, namely Dga1, Lro1, Are1 and Are2, and each of these enzymes show substrate promiscuity. Indeed, the deletion mutants of these acyltransferases show accumulation of DAGs across chain-lengths (Rockenfeller et al., 2018). So, there may be a mechanism by which these acyltransferases can achieve specificity or an unknown hitherto acyltransferase in yeast that has high substrate specificity. Considering the extent of dissection of lipid metabolic pathways in yeast and based on the available evidence on promiscuity of known acyltransferases, a high-fidelity conversion of C36:0 and C36:1 DAG to TAG is highly unlikely through any such means.

This leaves us with a possibility that DIP2 is an enzyme that drives specific DAG to TAG conversion. We have not been able to demonstrate it *in vitro* (formation of acyl-adenylate intermediate or TAG formation) using the purified protein. We believe that multiple parameters could have affected the ability of the protein. Lack of a partner (macromolecules or small organic molecules) or the presence of a bound inhibitor or the absence of a key modification or requiring the presentation of the substrate in a specific environment can be some of the reasons. The *in vitro* reaction will thus require a thorough investigation to optimize multiple variables, and therefore, will be the part of a future study. Nevertheless, the evidence presented here clearly points to the requirement of a unique acyltransferase as the existing acyltransferase are incapable of driving such high-fidelity DAG to TAG formation. Moreover, the lipidomics data using DIP2 carrying mutation of residue in the proposed active site, which is deeply buried in the protein, enables us to favour the proposal that DIP2 is a specific DAG to TAG converting enzyme.

A short summary of the points discussed is included in the discussion to the effect that ‘our studies do not rule out the possibility of an indirect role of DIP2 in DAG to TAG conversion’.

2) The authors should provide stronger evidence for the role of DIP2 in vacuoles.– Provide better images for the localization of DIP2 (see Rev. #1, point #1; Rev. #2; Rev. #3, point #4).

We thank the reviewer for their suggestion to improve the quality of images to unambiguously ascertain the localization. We concur with all the reviewers that the protein primarily localizes to mitochondria. In fact, previous studies have highlighted the localization to mitochondria in mouse brain tissue and NLT cell line and showed that DIP2A enables superoxide dismutase-mediated antioxidative response (Bai et al., 2021; Ma et al., 2019). Given that such an observation was already made which also included a thorough investigation, we focussed our attention on the secondary site of localization in yeast, i.e., vacuoles, which was not previously noted.

As per the suggestion by Reviewer 1, we have now performed the localization studies to generate better quality images. In addition, we have included a quantification of the fluorescence puncta colocalized with vacuolar and mitochondrial markers, FM4-64 and MitoTracker, respectively.

We find that ScDIP2, under normal growth conditions, shows 70-80% localization to mitochondria and 30-40 % to vacuoles. This observation not only supports the finding that DIP2 is localized to vacuoles but also highlights the fact that the localization of DIP2 to vacuole in yeast may have functional significance. It also shows that the localization of DIP2 may be a dynamic phenomenon that changes with stress and growth conditions. Indeed, while a recent study showed its localization to mitochondria in brain tissue, few other studies have reported its localisation to cell periphery and nucleus in mammalian cells (Hayashi et al., 2017; Ouchi et al., 2010). In *Drosophila*, DIP2 was found to be associated with a plasma membrane marker (Nitta et al., 2017).

The microscopy images now replace Figure 5A and 5B; Figure 5—figure supplement 1A-D in revised manuscript. The colocalization quantification data presented is included as Figure 5—figure supplement 1F and 1G in the revised manuscript. A summary of the observations made is now added to the Results section and also in the Discussion section at line# 251-267.

– Further test the role of DIP2 in vacuoles (is it direct or indirect?) (see Rev. #1, point #2and2b).

We sincerely thank the reviewer for suggesting experiments to further validate the importance of DIP2 localization to vacuole.

We used the point mutants of ScDIP2, where a conserved residue responsible for catalysis has been mutated and overexpressed in wildtype cells. We performed the quantification of vacuolar morphology to see if the vacuolar defect phenotype persists. As explained in the earlier question, the mutation will not impact any kind of protein interactions and therefore its effects will be restricted to the imbalances in DAG-TAG levels. Similar to deletion mutant, we find that this point mutation that prevents ScDIP2s ability to prevent specific DAG to TAG conversion, also has defective vacuolar fusion. It is to be noted that this observation supports the proposition that DIP2 is a specific enzyme driving selective DAG to TAG conversion.

The additional data are included as Figure 7—figure supplement 2F and 2G and a short summary of the points discussed is included in the manuscript (line# 335-337)

As an optional possibility, it would also be interesting to test the impact of DIP2 loss in LD biogenesis and shape.

Deletion of both the major TAG synthesising enzymes in yeast, Dga1 and Lro1 leads to more than 80% depletion of total TAG amount during logarithmic growth phase (Sorger and Daum, 2002). The effect of such TAG depletion is also reflected in LD biogenesis as the LD count per cell decreases from ~4 to almost null (Iwasa et al., 2016). However, as described in the manuscript, ScDIP2 works on minor subspecies of DAGs and facilitate their conversion to TAG subspecies. Therefore, we observed slight depletion (~17%) of total TAG pool (as shown in Figure 1—figure supplement 2D) upon ScDIP2 deletion in yeast at logarithmic phase and no significant change at stationary phase. Also, sterol is stored as sterol ester in LD and makes up 30-40% of LD mass. In the absence of ScDIP2, we did not observe any change in sterol ester and its precursor level (as shown in Figure 1—figure supplement 2D). Therefore, based on this evidence, we propose that ScDIP2 deletion might have minimal effect on LD biogenesis quantitatively.

However, we cannot rule out the possibility on the regulation aspect of LD biogenesis where excess accumulation of DAG subspecies might play some role. Though the experimentations towards this fall beyond the scope of the current manuscript, it certainly can be explored in future studies. A short summary of the results has been added to the results and Discussion sections (line# 409-412) in the revised manuscript.

Moreover, going further into the mechanism by which DIP2 has a specific role only in a subset of DAGs can be very interesting, but I feel that perhaps falls a bit beyond the scope of this paper. The authors can of course discuss it a bit further.

We thank the reviewer for the above suggestion. As mentioned in the answer to essential revision question#1, we need to first demonstrate the *in vitro* ability of the enzyme to convert DAG to TAG. Following these experiments, a thorough structural and biochemical investigation is required to further characterise the specificity of these enzymes which will be a focus of our future study.

Reviewer #1 (Recommendations for the authors):I see two weak points in this manuscript:1) The fluorescence microscopy data showing localization of the DIP2 protein to vacuole/mitochondria could be very much improved.

We thank the reviewer for the suggestion, and it has been answered as a part of the answers to essential revision#2. Improved microscopy images for ScDIP2 colocalization study is now included. Additionally, we have quantified the colocalization signal using the images. This is now presented as Figure 5A and 5B, Figure 5—figure supplement 1A-D and Figure 5—figure supplement 1F and 1G and discussed in the manuscript.

2) The vacuole fusion defect in the absence of DIP2 is quite interesting and promising. However, I feel like further data would be required to fully support such a claim. For instance, could the authors acutely remove DIP2 (anchor-away, degron system, etc.) to then perform time-lapse imaging to show, as predicted by the authors, vacuole fusion?

We thank the reviewer for the suggestion and this question is partly answered as a part of the answers to essential revision#2. Briefly, we overexpressed ScDIP2 carrying a point mutation that renders it non-functional as supporting evidence for the role of DIP2-mediated DAG regulation in vacuole fusion process. The overexpression of this ScDIP2 point mutant neither result in cell death under hypotonic conditions nor adversely impact vacuole fusion. This is now presented as Figure 7—figure supplement 2F and 2G and discussed in the manuscript.

(2b) Related to 1 and 2, would the authors have a way to monitor the lipid changes in specific organelles, such as vacuole or mitochondria?

We greatly appreciate the reviewer suggestion to measure lipid changes after isolating organelles. As revealed from the lipidomics experiments, the DAGs regulated by DIP2 only represent a tiny pool of the total lipidome (< 0.005%). Their abundance in the organelles and the quality of the signal from such rare pools is expected to be weak. Therefore, we will need to optimize conditions such as osmolarity or ER-stress to enhance the abundance of these rare species in the isolated organelles, which will be a focus of our future investigations.

3) Is DIP2 indeed a lipid metabolizing enzyme or is it a regulator/modifier of other metabolizing enzymes? I appreciate that the authors claim and write in their manuscript that they have not been able to show *in vitro* the capacity of DIP2 to convert some DAG species into TAG. How would the authors interpret such negative results? Because of experimental/technical limitations or because indeed DIP2 is not able to convert DAG *in vitro*?

We thank the reviewer for allowing us to expand on the alternative possibilities and interpret our failure to show *in vitro* DAG to TAG conversion. We have now elaborately discussed various possibilities and how evidence, though indirect as correctly pointed out, favours DIP2 facilitating DAG to TAG conversion as a part of essential revision#1.

Reviewer #2 (Recommendations for the authors):The authors should provide *in vitro* evidence of DIP2 activity (using recombinant proteins encompassing or not the FLDs and their mutants) and address the issue of its specificity at the molecular detail.

We agree with the reviewers’ concern that in the absence of *in vitro* evidence of DIP2 it is difficult to ascertain any specific role to DIP2. We have addressed these queries as a part of the essential revision#1. As part of the future study, we need to first establish a robust *in vitro* assay for DIP2 and then a thorough structural and biochemical investigation to delineate the specificity at the molecular level.

The localization of ScDIP2 is reported to be vacuolar + mitochondrial. The images shown by the authors would point towards a primarily mitochondrial localization. This aspect needs to be investigated further (and possibly expanded to mammalian and DmDIP2) in light of the proposed DIP2 biology.

We agree with the reviewers’ remark that localization is primarily to Mitochondria. However, mitochondrial localization was already discussed in an earlier study on mammalian and *Drosophila* systems and therefore we focussed on the vacuolar localization aspect. We have elaborately detailed this aspect as a part of the essential revision**#**2.

Reviewer #3 (Recommendations for the authors):1) As mentioned above, some DAG lipids go up in ScDIP2-ko. What is the effect on precursor lipids that make DAG, such as PA and lyso PA? Changes in these lipids can also influence ER homeostasis and ER stress.

We greatly appreciate the reviewer suggestion to rule out the possibility that difference in specific precursors have not caused the ER stress. We have addressed this as a part of the essential revision#1. This is now presented as Figure 2—figure supplement 1G and 1I and discussed in the manuscript.

2) The decrease in TAG in the ScDIP2 knockout could be due to an increase in lipolysis or a decrease in TAG synthesis. Here, increased lipolysis is ruled out because there is no significant increase in MAG or FFAs in this KO (Figure 2, figure supplement 2B,c). This conclusion would be more strongly supported if the rate of glycerol release could be monitored in control and ScDIP2 KO. MAG and FFAs can often be rapidly re-incorporated into other lipid species, or simply oxidized. Thus, the lack of their accumulation alone does not seem sufficient to rule out elevated lipolysis in the KO.

We greatly appreciate the reviewer’s concern that without measuring glycerol levels alongside FFA and MAGs, one cannot completely rule out lipolysis dependent DAG-TAG imbalance. We have addressed this issue as a part of the essential revision**#1** and modified the manuscript text as “to rule out completely the role of lipolysis, glycerol level also needs to be measured.”

3) Loss of ScDIP2 induces ER UPR stress, which can be rescued by treatment with oleic acid (Figure 3C). Oleic acid treatment drives lipid droplet biogenesis in yeast, which is commonly associated with releasing various ER stresses. Does this rescue require LD biogenesis? Relatedly, how does loss of ScDIP2 impact LD biogenesis and morphology?

We thank the reviewer to point us to the potential link between oleic acid (OL) driven LD biogenesis and ER stress modulation. OL treatment, in general, is known to increase TAG synthesis which subsequently leads to LD biogenesis. Previous studies revealed that the OL treatment promotes TAG production and thereby relieves the ER stress by quenching the toxic saturated lipids in form of TAGs (Volmer and Ron, 2015). We also observe a surge in TAG synthesis (~ 3-7-fold) in both wildtype and ScDIP2 null yeast upon addition of OL, suggesting a subsequent increase in LD biogenesis.

It should be emphasized here that this DAG to TAG conversion upon addition of OL is indiscriminate. Therefore, the accumulated toxic DAG subspecies are quenched alongside other DAGs relieving the cell from ER stress. It is evident from the lipidomic data where an upregulation of TAG synthesis along with buffering of all DAG species including C36:0 and C36:1 DAG. This is observed in two cases; in stationary phase (Figure 4A) when TAG production is naturally triggered or when mutant is treated with OL (Figure 3 E and 3F). We also have shown that ER stress is reduced in ScDIP2 null yeast during stationary phase as well as upon treatment with OL.

We have added this additional data as Figure 3—figure supplement 2G in the manuscript (line# 215-217).

4) Through primarily co-imaging with ScDIP2-GFP and vacuole strain FM4-64, it is proposed that a pool of SCDIP2-GFP localizes to the vacuole surface. As presented in Figure 5A, this ScDIP2-GFP does not look vacuolar, but possibly localized to mitochondria adjacent to the vacuole. Some additional evidence supporting vacuole localization would strengthen this observation. Potentially blotting for ScDIP2 on isolated vacuoles.

We agree with the reviewers’ remark that localization is primarily to mitochondria. We have elaborately detailed this aspect as a part of the essential revision #2. We have now provided a quantification of the colocalization signal between ScDIP2-GFP puncta and organellar markers (mitochondria and vacuole) signal.

In the earlier version of the manuscript, we have shown the association of ScDIP2 with total membrane fraction and not in cytosolic fraction via western blotting experiment (Figure 5—figure supplement 1E). Although the signal was weak, it was indicating its presence in membrane fraction. The weak signal can be attributed to the low endogenous expression levels of DIP2 in yeast (~ 41 molecules per cell in log phase) (Ghaemmaghami et al., 2003). Indeed, when we attempted to probe ScDIP2 in isolated vacuoles from yeast, its low abundance hindered the detection of signal in western blot. It may require certain conditions or optimisation to enrich ScDIP2 on vacuolar surface to get detectable signal from western blot experiments.

References:

Bai, L. L., Zhang, L. Q., Ma, J., Li, J., Tian, M., Cao, R. J., He, X. X., He, Z. X., Yu, H. L., and Zhu, X. J. (2021). DIP2A is involved in SOD-mediated antioxidative reactions in murine brain. *Free Radic Biol Med*, *168*, 6-15. https://doi.org/10.1016/j.freeradbiomed.2021.03.027

Casanovas, A., Sprenger, R. R., Tarasov, K., Ruckerbauer, D. E., Hannibal-Bach, H. K., Zanghellini, J., Jensen, O. N., and Ejsing, C. S. (2015). Quantitative analysis of proteome and lipidome dynamics reveals functional regulation of global lipid metabolism. *Chem Biol*, *22*(3), 412-425. https://doi.org/10.1016/j.chembiol.2015.02.007

Chakravarty, S., and Varadarajan, R. (1999). Residue depth: a novel parameter for the analysis of protein structure and stability. *Structure*, *7*(7), 723-732. https://doi.org/10.1016/s0969-2126(99)80097-5

Ejsing, C. S., Sampaio, J. L., Surendranath, V., Duchoslav, E., Ekroos, K., Klemm, R. W., Simons, K., and Shevchenko, A. (2009). Global analysis of the yeast lipidome by quantitative shotgun mass spectrometry. *Proc Natl Acad Sci U S A*, *106*(7), 2136-2141. https://doi.org/10.1073/pnas.0811700106

Elbaz-Alon, Y., Rosenfeld-Gur, E., Shinder, V., Futerman, A. H., Geiger, T., and Schuldiner, M. (2014). A dynamic interface between vacuoles and mitochondria in yeast. *Dev Cell*, *30*(1), 95-102. https://doi.org/10.1016/j.devcel.2014.06.007

Ghaemmaghami, S., Huh, W. K., Bower, K., Howson, R. W., Belle, A., Dephoure, N., O'Shea, E. K., and Weissman, J. S. (2003). Global analysis of protein expression in yeast. *Nature*, *425*(6959), 737-741. https://doi.org/10.1038/nature02046

Hayashi, T., Lombaert, I. M., Hauser, B. R., Patel, V. N., and Hoffman, M. P. (2017). Exosomal MicroRNA Transport from Salivary Mesenchyme Regulates Epithelial Progenitor Expansion during Organogenesis. *Dev Cell*, *40*(1), 95-103. https://doi.org/10.1016/j.devcel.2016.12.001

Heier, C., Taschler, U., Rengachari, S., Oberer, M., Wolinski, H., Natter, K., Kohlwein, S. D., Leber, R., and Zimmermann, R. (2010). Identification of Yju3p as functional orthologue of mammalian monoglyceride lipase in the yeast *Saccharomyces cerevisiae*. *Biochim Biophys Acta*, *1801*(9), 1063-1071. https://doi.org/10.1016/j.bbalip.2010.06.001

Iwasa, S., Sato, N., Wang, C. W., Cheng, Y. H., Irokawa, H., Hwang, G. W., Naganuma, A., and Kuge, S. (2016). The Phospholipid:Diacylglycerol Acyltransferase Lro1 Is Responsible for Hepatitis C Virus Core-Induced Lipid Droplet Formation in a Yeast Model System. *PLoS One*, *11*(7), e0159324. https://doi.org/10.1371/journal.pone.0159324

Jumper, J., Evans, R., Pritzel, A., Green, T., Figurnov, M., Ronneberger, O., Tunyasuvunakool, K., Bates, R., Zidek, A., Potapenko, A., Bridgland, A., Meyer, C., Kohl, S. A. A., Ballard, A. J., Cowie, A., Romera-Paredes, B., Nikolov, S., Jain, R., Adler, J.,... Hassabis, D. (2021). Highly accurate protein structure prediction with AlphaFold. *Nature*, *596*(7873), 583-589. https://doi.org/10.1038/s41586-021-03819-2

Kurat, C. F., Natter, K., Petschnigg, J., Wolinski, H., Scheuringer, K., Scholz, H., Zimmermann, R., Leber, R., Zechner, R., and Kohlwein, S. D. (2006). Obese yeast: triglyceride lipolysis is functionally conserved from mammals to yeast. *J Biol Chem*, *281*(1), 491-500. https://doi.org/10.1074/jbc.M508414200

Ma, J., Chen, L., He, X. X., Wang, Y. J., Yu, H. L., He, Z. X., Zhang, L. Q., Zheng, Y. W., and Zhu, X. J. (2019). Functional prediction and characterization of Dip2 gene in mice. *Cell Biol Int*, *43*(4), 421-428. https://doi.org/10.1002/cbin.11106

Nitta, Y., Yamazaki, D., Sugie, A., Hiroi, M., and Tabata, T. (2017). DISCO Interacting Protein 2 regulates axonal bifurcation and guidance of *Drosophila* mushroom body neurons. *Dev Biol*, *421*(2), 233-244. https://doi.org/10.1016/j.ydbio.2016.11.015

Oelkers, P., Cromley, D., Padamsee, M., Billheimer, J. T., and Sturley, S. L. (2002). The DGA1 gene determines a second triglyceride synthetic pathway in yeast. *J Biol Chem*, *277*(11), 8877-8881. https://doi.org/10.1074/jbc.M111646200

Ouchi, N., Asaumi, Y., Ohashi, K., Higuchi, A., Sono-Romanelli, S., Oshima, Y., and Walsh, K. (2010). DIP2A functions as a FSTL1 receptor. *J Biol Chem*, *285*(10), 7127-7134. https://doi.org/10.1074/jbc.M109.069468

Patil, G. S., Kinatukara, P., Mondal, S., Shambhavi, S., Patel, K. D., Pramanik, S., Dubey, N., Narasimhan, S., Madduri, M. K., Pal, B., Gokhale, R. S., and Sankaranarayanan, R. (2021). A universal pocket in Fatty acyl-AMP ligases ensures redirection of fatty acid pool away from Coenzyme A-based activation. *ELife*, *10*. https://doi.org/10.7554/*eLife*.70067

Rockenfeller, P., Smolnig, M., Diessl, J., Bashir, M., Schmiedhofer, V., Knittelfelder, O., Ring, J., Franz, J., Foessl, I., Khan, M. J., Rost, R., Graier, W. F., Kroemer, G., Zimmermann, A., Carmona-Gutierrez, D., Eisenberg, T., Buttner, S., Sigrist, S. J., Kuhnlein, R. P.,... Madeo, F. (2018). Diacylglycerol triggers Rim101 pathway-dependent necrosis in yeast: a model for lipotoxicity. *Cell Death Differ*, *25*(4), 767-783. https://doi.org/10.1038/s41418-017-0014-2

Schweiger, M., Eichmann, T. O., Taschler, U., Zimmermann, R., Zechner, R., and Lass, A. (2014). Measurement of lipolysis. *Methods Enzymol*, *538*, 171-193. https://doi.org/10.1016/B978-0-12-800280-3.00010-4

Selvaraju, K., Gowsalya, R., Vijayakumar, R., and Nachiappan, V. (2016). MGL2/YMR210w encodes a monoacylglycerol lipase in *Saccharomyces cerevisiae*. *FEBS Lett*, *590*(8), 1174-1186. https://doi.org/10.1002/1873-3468.12136

Sorger, D., and Daum, G. (2002). Synthesis of triacylglycerols by the acyl-coenzyme A:diacyl-glycerol acyltransferase Dga1p in lipid particles of the yeast *Saccharomyces cerevisiae*. *J Bacteriol*, *184*(2), 519-524. https://doi.org/10.1128/JB.184.2.519-524.2002

Tan, K. P., Nguyen, T. B., Patel, S., Varadarajan, R., and Madhusudhan, M. S. (2013). Depth: a web server to compute depth, cavity sizes, detect potential small-molecule ligand-binding cavities and predict the pKa of ionizable residues in proteins. *Nucleic Acids Res*, *41*(Web Server issue), W314-321. https://doi.org/10.1093/nar/gkt503

[Editors' note: further revisions were suggested prior to acceptance, as described below.]

The manuscript has been improved but there are some remaining issues that need to be addressed, as outlined below:We all agree that this study is generally well done and that it constitutes a potentially exciting piece of work. However, upon consultation amongst reviewers, we still think that the interpretation of the new microscopy data as a proof for vacuolar localization of ScDIP2 is not conclusive. An alternative explanation of the data could be that all ScDIP2 localizes to mitochondria, and that the vacuole association (as observed by fluorescence microscopy) represents mitochondrial-vacuole contacts.In order to clarify better this point, we request that you perform 3-color imaging of DIP2 together with mitochondrial and vacuolar markers, which will help reveal whether the association between DIP2 and vacuoles as observed in the present version of the manuscript corresponds to mitochondrial-vacuolar contacts or not. These data could even provide an exciting new observation if it confirms mitochondria-vacuole contacts, since there are not that many proteins known to localize to these contacts. The authors should adjust the text according to the outcome of these experiments.

We sincerely thank the reviewer and editor for the excellent suggestion to probe if the observation of ScDIP2-vacuolar association corresponds to a mitochondria-vacuole contact sites. As suggested, a three colour colocalization experiment was performed using labelled ScDIP2 (GFP tagging, Green fluorescent protein), Mitochondria (DNA staining; DAPI) and Vacuole (Membrane; FM4-64). As indicated by both the reviewer and the editor, it indeed represents the site of contact between mitochondria-vacuoles. This observation has added a new dimension to the findings and allows us to say that DIP2 is a new family of protein that localizes to mitochondria-vacuole contact site. We once again sincerely thank the reviewer for pointing us to probe in this direction, which has significantly improved the quality of the manuscript.

These findings are now shown as Figure 5A, 5B and Figure 5—figure supplement 1H and 1I in the revised manuscript. It is also now mentioned in the main text (line number 21-22; line number 73-75; line number 261-274 and 276-277). These new findings and its implications are also discussed briefly in the discussion (line number 408-421). Key references for these experiments have also been included (References- 21, 33, 42, 46, 49, 98 and 119). The details of this microscopy experiment is included in the materials and method section (line number 654-659).

Reviewer #3 (Recommendations for the authors):The study investigates the role of DIP2 enzymes primarily in yeast, and their roles in neutral lipid metabolism. In general the study is well conducted and provides significant preliminary data on the role of these enzymes in DAG:TAG lipid metabolism. The only major remaining concern is the conclusion of the vacuole localization of ScDIP2. Genetic manipulation of ScDIP2 clearly affects vacuole morphology and fusion, etc, but the fluorescent imaging data, as presented, does not conclusively show vacuole localization of ScDIP2-GFP. An alternative model is that all ScDIP2 simply localizes to mitochondria, and that the vacuole association simply represents mito-vacuole contacts. The new imaging data presented in this revised manuscript are compelling, but cannot rule out this alternative model. Since this vacuole localization appears to be a major tenet of the study's conclusions, additional evidence such as biochemical analysis, or even imaging with dual mito and vacuole markers, is necessary prior to making such conclusions.

We sincerely thank reviewer-3 for the suggestion to carry out imaging with dual mitochondria and vacuole markers. ScDIP2 indeed localizes to mitochondria-vacuole contact sites. The suggestion has definitively added a new flavour to the story. The query is addressed in detail in the essential revision query of the editor. Accordingly, changes have been made to Figure-5 and related supplementary files, main text and Discussion sections.